# *Mir221/222* drive synovial hyperplasia and arthritis by targeting cell cycle inhibitors and chromatin remodeling components

**Fani Roumelioti**[1,2], **Christos Tzaferis**[1,3], **Dimitris Konstantopoulos**[1], **Dimitra Papadopoulou**[1,3], **Alejandro Prados**[1], **Maria Sakkou**[1,4], **Anastasios Liakos**[5], **Panagiotis Chouvardas**[1], **Theodore Meletakos**[5], **Yiannis Pandis**[1], **Niki Karagianni**[6], **Maria C Denis**[6], **Maria Fousteri**[5], **Maria Armaka**[5], **George Kollias**[1,3,4]*

[1]Institute for Bioinnovation, Biomedical Sciences Research Centre (BSRC) "Alexander Fleming", Vari, Greece; [2]Department of Pathophysiology, Medical School, National and Kapodistrian University of Athens, Athens, Greece; [3]Department of Physiology, Medical School, National and Kapodistrian University of Athens, Athens, Greece; [4]Center of New Biotechnologies & Precision Medicine, National and Kapodistrian University of Athens Medical School, Athens, Greece; [5]Institute for Fundamental Biomedical Research, Biomedical Sciences Research Center "Alexander Fleming", Vari, Greece; [6]Biomedcode Hellas SA, Vari, Greece

**Abstract** miRNAs constitute fine-tuners of gene expression and are implicated in a variety of diseases spanning from inflammation to cancer. miRNA expression is deregulated in rheumatoid arthritis (RA); however, their specific role in key arthritogenic cells such as the synovial fibroblast (SF) remains elusive. Previous studies have shown that *Mir221/222* expression is upregulated in RA SFs. Here, we demonstrate that TNF and IL-1β but not IFN-γ activated *Mir221/222* gene expression in murine SFs. SF-specific overexpression of *Mir221/222* in huTNFtg mice led to further expansion of SFs and disease exacerbation, while its total ablation led to reduced SF expansion and attenuated disease. *Mir221/222* overexpression altered the SF transcriptional profile igniting pathways involved in cell cycle and ECM (extracellular matrix) regulation. Validation of targets of *Mir221/222* revealed cell cycle inhibitors *Cdkn1b* and *Cdkn1c*, as well as the epigenetic regulator *Smarca1*. Single-cell ATAC-seq data analysis revealed increased *Mir221/222* gene activity in pathogenic SF subclusters and transcriptional regulation by *Rela*, *Relb*, *Junb*, *Bach1*, and *Nfe2l2*. Our results establish an SF-specific pathogenic role of *Mir221/222* in arthritis and suggest that its therapeutic targeting in specific subpopulations could lead to novel fibroblast-targeted therapies.

*For correspondence:
kollias@fleming.gr

## Editor's evaluation

The findings of this work are important and offer significant advances to current knowledge. This manuscript used state of the art techniques and employed relevant animal models to provide convincing evidence supporting the regulatory role of microRNA cluster 221/222 in rheumatoid arthritis synovial fibroblast. Targeting miR-221/222 in SFs of patients harboring inflammatory arthritis might have a therapeutic benefit, and will be interesting to a wide range audience in the rheumatology and bone research fields.

## Introduction

Rheumatoid arthritis (RA) is a chronic, painful, and destructive disease with sustained inflammation in the joints, which eventually leads to the destruction of cartilage and bone. Articular inflammation is fueled by the aberrant production of inflammatory mediators by immune and mesenchymal resident cells (*McInnes et al., 2016*). Genetically modified mouse models with deregulated expression of TNF develop an erosive polyarthritis resembling RA (huTNFtg) (*Keffer et al., 1991*) and SpAs (spondyloarthropathies) (TNF$^{\Delta ARE}$ [*Kontoyiannis et al., 1999*], TgA86 [*Küsters et al., 1997*; *Vafeiadou et al., 2020*]), showing also extra-articular manifestations, such as heart valve disease and intestinal inflammation, recapitulating comorbid pathologies often observed in humans (*Kontoyiannis et al., 1999*; *Ntari et al., 2018*).

Early studies in our lab established that TNF targeting of TNFR1 on synovial fibroblasts (SFs) is required for the development of arthritis and suffices to orchestrate full pathogenesis (*Armaka et al., 2008*; *Armaka et al., 2018*). SFs (resident mesenchymal cells in the joints) in RA diverge from their physiological role to nourish and protect the joints and undergo arthritogenic transformation. Under chronic inflammatory signals, SFs hyperproliferate, express ECM-degrading enzymes, and acquire an aggressive, invasive, and tissue-destructive phenotype (*Kontoyiannis and Kollias, 2000*; *Dakin et al., 2018*; *Koliaraki et al., 2020*). In addition, arthritogenic SFs can migrate via the circulation to distant sites and transfer disease in both mice and humans (*Aidinis et al., 2003*; *Lefèvre et al., 2009*). Epigenetic alterations, such as global DNA hypomethylation and deregulated expression of miRNAs and lncRNAs (*Karouzakis et al., 2009*; *Evangelatos et al., 2019*; *Zou et al., 2018*), have been linked to the aggressive and destructive behavior of SFs.

Altered expression and important functions have been reported for a number of miRNAs in RA (*Evangelatos et al., 2019*). miRNAs are small regulatory RNAs (19–25 nt long) that fine-tune the expression of target mRNAs and have fundamental roles in development, physiology, and disease. They recognize their RNA targets by binding mainly to their 3′UTR region, guide them to the RISC complex, and lead them either to target degradation or translational repression (*Jonas and Izaurralde, 2015*).

Specifically, *Mir221* and *Mir222* have been found upregulated in SFs derived from the huTNFtg mouse model and RA patients (*Pandis et al., 2012*). *Mir221/222* are clustered on the X chromosome (*Mirc53* locus) and classified as oncomiRs as they are upregulated in a number of human epithelial cancers. They target genes mainly involved in cell proliferation and survival, such as cell cycle inhibitors *Cdkn1b* and *Cdkn1c* (*le Sage et al., 2007*; *Pineau et al., 2010*). Under nonresolving inflammatory signals, such as in sepsis, the overexpression of *Mir222* contributes to morbidity by targeting *Smarca4*, a chromatin remodeling component that participates in the SWI/SNF complex (*Seeley et al., 2018*). In RA, expression levels of *Mir221/222* positively correlate with disease progression (*Abo ElAtta et al., 2019*) and ex vivo downregulation of *Mir221* has been shown to decrease migration and invasion of human RA SFs (*Yang and Yang, 2015*). However, the in vivo role of these two miRNAs in arthritis remained unexplored.

Here, we provide evidence that *Mir221/222* play a key pathogenic role in TNF-driven arthritis as in vivo SF-targeted overexpression of *Mir221/222* led to more aggressive arthritis associated with enhanced expansion of SF populations and total genetic ablation of these miRNAs led to amelioration of disease manifestations. Additionally, bioinformatic target prediction tools for *Mir221/222* combined with RNA expression data uncovered both known [*Cdkn1b (p27), Cdkn1c (p57)* cell cycle inhibitors] as well as novel targets (*Smarca1*, a chromatin remodeling component). scATAC-seq analysis of SFs from arthritic mice revealed increased *Mir221/222* gene activity in the destructive subpopulations of the lining and expanding intermediate compartment. Finally, transcription factor motif enrichment analysis revealed transcription factors such as *Rela, Relb, Junb* (*Galardi et al., 2011*), *Nfe2l2*, and *Bach1* that could regulate the expression of *Mir221/222* in arthritis.

Overall, these data suggest that therapeutic targeting of these two miRNAs in specific SF subpopulations could be beneficial for the treatment of human arthritis and perhaps other inflammatory diseases associated with pathogenic expansion of fibroblasts.

# Results

## *Mir221* and *Mir222* induction in huTNFtg SFs depends on the TNF/TNFR1 axis and IL-1β

We have previously shown that *Mir221/222* levels are increased in SFs from arthritic 8-week-old huTNFtg mice and RA patients (*Pandis et al., 2012*). To determine the expression profile of both *Mir221* and *Mir222* during disease progression, we analyzed their expression in huTNFtg SFs obtained at early (3 weeks), intermediate (8 weeks), as well as late (11 weeks) disease stages, and compared them to SFs from WT control mice. Both *Mir221* and *Mir222* were upregulated from the early disease stage, and this upregulation was further augmented with disease progression (*Figure 1A and B*).

Previous studies have shown that *Mir221/222* expression is induced by LPS and TNF in macrophages and that they are implicated in LPS tolerance and endotoxemia (*Seeley et al., 2018*; *El Gazzar and McCall, 2010*). To understand the tissue- and disease-specific regulation of *Mir221/222* in SFs, we stimulated SFs with an array of inflammatory stimuli, such as TNF, LPS, IL-1β, IFN-γ, and PolyIC, and analyzed miRNA expression 24, 48, and 72 hr post-induction (*Figure 1C and D*). To validate the response of SFs to the inflammatory signals, we measured the expression of known downstream targets (*Figure 1—figure supplement 1A and B*). TNF, LPS, and IL-1β led to consistent upregulation of both *Mir221* and *Mir222* in SFs at all time points examined (*Figure 1C and D*). To further characterize the requirement of upstream TNF signaling for the induction of *Mir221/222*, SFs were isolated from huTNFtg mice lacking TNFR1, and *Mir221/222* levels were measured. These mice do not develop any RA pathology and the two miRNAs were found not to be induced in the absence of TNFR1 (*Figure 1E and F*). Additionally, we analyzed whether the TNF-mediated induction of *Mir221/222* depends on IL-1β signaling. Since IL-1β signaling is reported to act both independently or downstream of TNF in RA and SFs (*Brennan et al., 1989*; *Probert et al., 1995*; *van den Berg et al., 1999*; *Feldmann, 2002*; *Zwerina et al., 2007*), we treated WT SFs with anakinra (an IL-1 receptor antagonist) and then stimulated them with TNF. Anakinra successfully inhibited downstream IL-1β signaling (*Figure 1—figure supplement 1C*), but did not abrogate *Mir221/222* induction by TNF (*Figure 1G and H*). Thus, TNF may upregulate *Mir221/222* independently of IL-1β.

## Mesenchymal cell overexpression of *Mir221* and *Mir222* exacerbates arthritis in huTNFtg mice

In order to study the in vivo role of *Mir221/222* in SFs, we generated a transgenic mouse, TgCol6a1-Mir221/222, overexpressing these two miRNAs under the *Col6a1* promoter (*Figure 2A*), which is known to target cells of mesenchymal origin in the joints, including the SFs (*Armaka et al., 2008*). Overexpression of *Mir221/222* was verified through qRT-PCR analysis in various tissues (*Figure 2—figure supplement 1A and B*). Fibroblast-specific overexpression of *Mir221/222* was verified in ex vivo cell cultures of SFs, intestinal mesenchymal cells (IMCs), and lung fibroblasts (LFs) (*Figure 2—figure supplement 1C and D*), whereas no overexpression could be observed in cells of hematopoietic or epithelial lineage (*Figure 2—figure supplement 1E and F*).

To explore the role of the two miRNAs in the pathogenesis of arthritis, we crossed the TgCol6a1-Mir221/222 with the huTNFtg mice and monitored disease progression. Histopathological analysis revealed exacerbated arthritis, as observed by enhanced synovial hyperplasia (pannus formation), cartilage destruction, and number of osteoclasts in the double huTNFtg;TgCol6a1-Mir221/222 transgenic mice in comparison to the huTNFtg controls (*Figure 2B*). Moreover, bone morphometric analysis using microCT revealed greater bone erosions as indicated by decreased bone volume, decreased trabecular thickness, and increased trabecular separation (*Figure 2C–F*). Interestingly, comparison of TgCol6a1-Mir221/222 to WT mice revealed significantly decreased trabecular thickness, as well as a tendency for decreased bone volume and increased trabecular separation (*Figure 2D–F*, *Figure 2—figure supplement 2A*), suggesting a potential a priori function of the two miRNAs in bone physiology, which needs further elucidation. To examine whether the effect of *Mir221/222* on arthritis manifestations was SF-mediated and exclude a contributing indirect role of *Mir221/222* sufficiency from other cell types, we generated huTNFtg;TgCol6a1-Mir221/222;Mir221/222⁻/⁻ mice. Crossing mice carrying targeted alleles of *Mir221/222* with loxP sites (Mir221/222 ᶠ/ᶠ) with a Deleter-Cre (*Schwenk et al., 1995*) mouse line led to the generation of complete knockout mice, referred to as Mir221/222⁻/⁻. Deletion of *Mir221/222* in SFs of Mir221/222⁻/⁻ mice was confirmed by qRT-PCR (*Figure 2—figure*

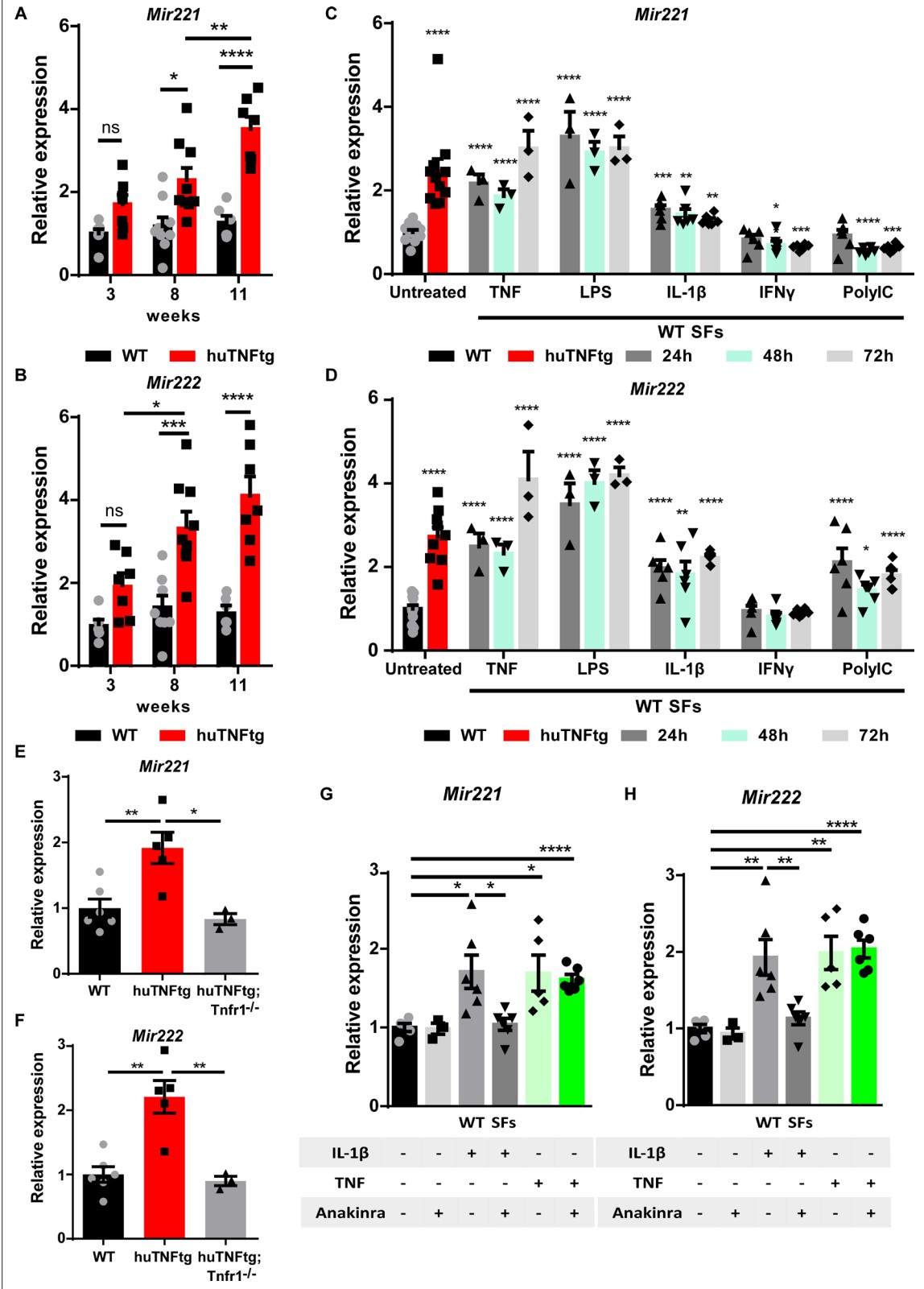

**Figure 1.** Regulation of *Mir221/222* levels in synovial fibroblasts (SFs) under arthritogenic signals. (**A, B**) Expression levels of *Mir221* and *Mir222* from cultured SFs, as determined by qRT-PCR at different time points of disease of huTNFtg mice, as well as control wild-type mice (WT). Expression levels were normalized to the levels seen in 3-week-old WT mice (n = 6–9). Two-way ANOVA statistical analysis was used with the suggested correction. (**C, D**) Quantification of *Mir221* and *Mir222* levels in cultured WT SFs after 24, 48, and 72 hr stimulation with TNF, LPS, IL1-β, IFN-γ, and PolyIC (n = 3–13).

*Figure 1 continued on next page*

Figure 1 continued

WT unstimulated SFs served as reference. Multiple *t*-tests statistical analysis was used. (**E, F**) *Mir221* and *222* levels in cultured SFs from huTNFtg and huTNFtg;Tnfr1[-/-] mice (n = 3–6). WT SFs served as reference for normalization. Student's *t*-test statistical analysis was used, unpaired. (**G, H**) *Mir221* and *222* expression analysis in cultured WT SFs after TNF or IL-1β stimulation in the absence or presence of anakinra (n = 3–6). Expression was normalized to the levels detected in WT unstimulated SFs. Student's *t*-test statistical analysis was used, unpaired. In all expression analysis experiments for *Mir221* and *222*, *Rnu6* was used as a housekeeping gene. Data represent mean ± SEM. *p<0.05, **p<0.01, ***p<0.001, ****p<0,0001, ns = not significant.

The online version of this article includes the following figure supplement(s) for figure 1:

**Figure supplement 1.** Synovial fibroblasts (SFs) respond to inflammatory signals.

*supplement 2B, C*). Mir221/222[-/-] mice developed normally and did not exhibit any obvious phenotypic defects. huTNFtg;TgCol6a1-Mir221/222;Mir221/222[-/-] mice exhibited synovial hyperplasia, cartilage destruction, and number of osteoclasts as the huTNFtg;TgCol6a1-Mir221/222;Mir221/222 [d/f, f/-] mice (*Figure 2G*), indicating that SF-specific *Mir221/222* overexpression even in the absence of *Mir221/222* in other cell types is sufficient to deteriorate arthritis manifestations.

Next, we checked the role of *Mir221/222* overexpression in immune infiltration in the joints of huTNFtg;TgCol6a1-Mir221/222 mice and observed that it did not lead to a significant increase in inflammatory infiltrations, apart from an increase in CD8[+] T cells (*Figure 3A*, *Figure 3—figure supplement 1A and B*) compared to huTNFtg mice. CD8[+] T cell subpopulations were further characterized and effector CD8[+] T cells were found to be the predominant CD8[+] T cell subpopulation in the arthritic joints. Moreover, effector CD8[+] T cells were increased in the joints of huTNFtg;TgCol6a1-Mir221/222 compared to the huTNFtg mice and accounted for the observed increase in CD8[+] T cells (*Figure 3—figure supplement 1B*, *Figure 3—figure supplement 2A–D*).

To examine the effect of *Mir221/222* overexpression in the physiology of SFs, we focused on the analysis of known fibroblast subpopulations in the joints of the huTNFtg;TgCol6a1-Mir221/222 mice. Similar to human RA, both lining layer SFs (LLSFs) and sublining layer SFs (SLSFs) were expanded in huTNFtg mice compared to WT, as indicated by FACS analysis (*Figure 3B*, *Figure 3—figure supplement 1C*). Notably, *Mir221/222* overexpression in arthritis rendered both layers more hyperplastic in huTNFtg;TgCol6a1-Mir221/222 mice compared to huTNFtg. It is worth mentioning that *Mir221/222* overexpression alone was sufficient to increase the number of fibroblasts of the LL and SL in TgCol6a1-Mir221/222 mice (*Figure 3B*). To define which SF subpopulation responds to inflammatory signals produced by the arthritogenic microenvironment and upregulates *Mir221/222* expression, we sorted LL and SL fibroblasts and measured *Mir221/222* expression. An increase in *Mir221/222* levels was observed, mainly, in fibroblasts of the LL and to a lesser extent in those of the SL (*Figure 3C and D*).

Finally, primary SFs isolated from the joints of huTNFtg;TgCol6a1-Mir221/222 mice were functionally characterized ex vivo. In agreement with the in vivo data, both WT and huTNFtg SFs overexpressing *Mir221/222* exhibited increased proliferation and wound-healing potential compared to their respective controls (*Figure 3E and F*). Collectively, these results establish a pro-proliferative function of *Mir221/222* on SFs.

## Overexpression of *Mir221/222* in SFs results in activated pathways involved in cell cycle and repressed pathways linked to ECM remodeling

To identify downstream targets of *Mir221/222* and unveil the molecular events that mediate the effect of their overexpression, we performed RNA sequencing and analyzed the expression profile of SFs from WT, TgCol6a1-Mir221/222, huTNFtg, and huTNFtg;TgCol6a1-Mir221/222 mice. The different genotypes are presented with distinct expression signatures and grouped according to genotype (*Figure 4—figure supplement 1A*). We found a significant number of genes deregulated in all conditions compared to WT SFs, as well as an increase in deregulated genes due to *Mir221/222* overexpression in the arthritogenic environment of huTNFtg mice (*Figure 4—figure supplement 1B and C*). To assess the effect of *Mir221/222*, we focused on genes upregulated (n = 399) and downregulated (n = 449) in huTNFtg;TgCol6a1-Mir221/222 SFs compared to huTNFtg (*Figure 4A*). KEGG pathway analysis of significantly differentially expressed genes showed strong over-representation of pathways involved in cell cycle and DNA replication in the group of upregulated genes, while functional terms such as ECM function and AKT signaling were enriched among the downregulated ones (*Figure 4A*).

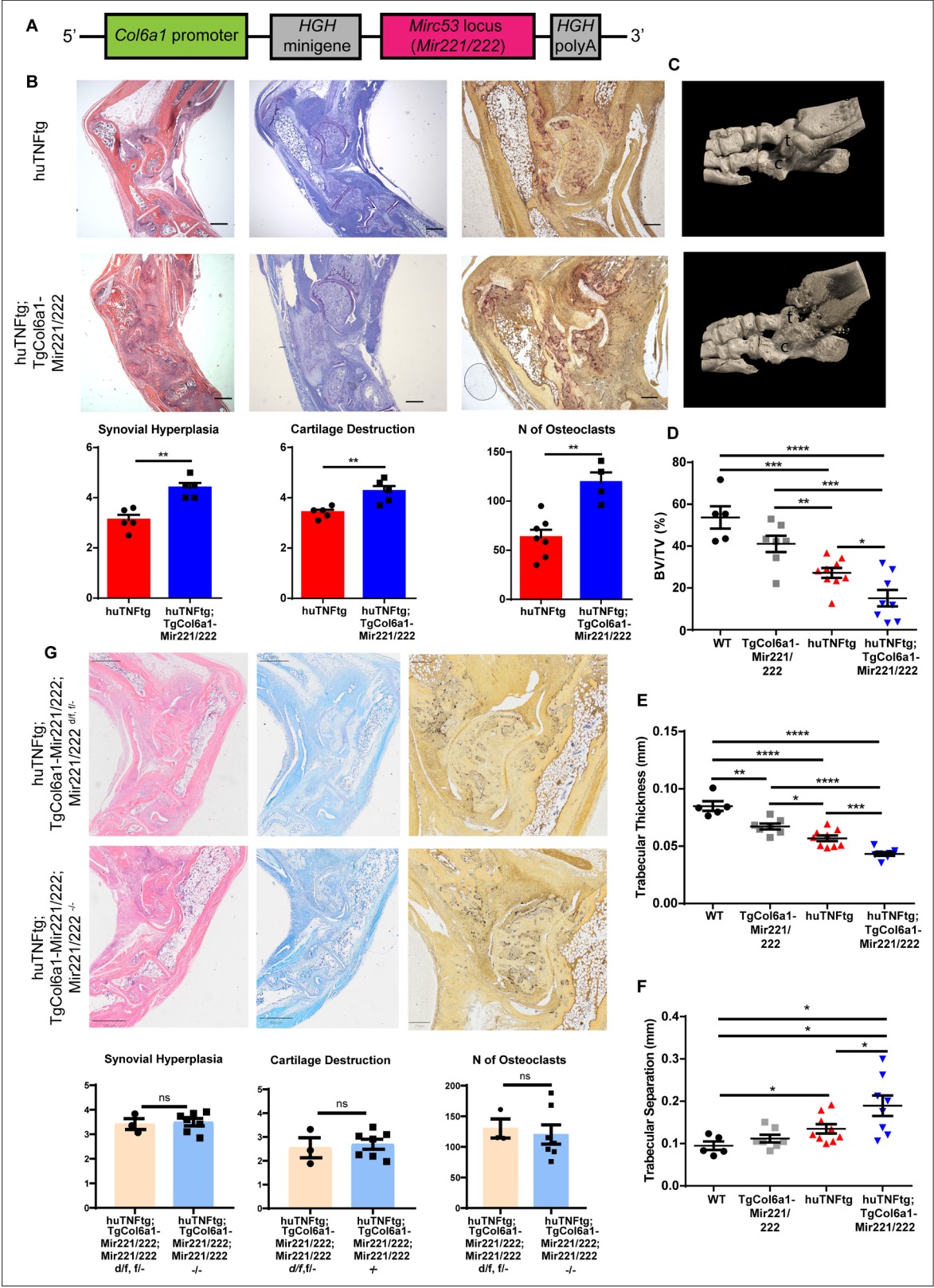

**Figure 2.** Mesenchymal *Mir221/222* overexpression leads to worse arthritis manifestations in huTNFtg mice. (**A**) *Mir221/222* were cloned under the *Col6α1* promoter to target expression in cells of mesenchymal origin. (**B**) Representative histological images of H&E, TB, and TRAP stained ankle joint sections and histological score of synovial hyperplasia, cartilage destruction, and osteoclast numbers of 8-week-old huTNFtg (n = 5–7) and huTNFtg;TgCol6a1-Mir221/222 mice (n = 4–5). t: talus; c: calcaneous. Scale bars: 600 μm and 300 μm. Student's *t*-test statistical analysis

*Figure 2 continued on next page*

Figure 2 continued

was used, unpaired. (**C**) Representative microCT images of the ankle joint area of 8-week-old huTNFtg and huTNFtg;TgCol6a1-Mir221/222 mice. (**D–F**) Quantification of bone erosions measuring the following parameters: decreased bone volume, decreased trabecular thickness, and increased trabecular separation by microCT analysis in the ankle joints of WT (n = 5), TgCol6a1-Mir221/222 (n = 7), huTNFtg (n = 9), and huTNFtg;TgCol6a1-Mir221/222 mice (n = 8). BV: bone volume; TV: trabecular volume. Student's *t*-test statistical analysis was used, unpaired. Data represent mean ± SEM. *p<0.05, **p<0.01, ***p<0.001, ****p<0,0001, ns = not significant. (**G**) Representative histological images of H&E, TB, and TRAP stained ankle joint sections and histological score of synovial hyperplasia, cartilage destruction, and osteoclast numbers of 7–8-week-old huTNFtg;TgCol6a1-Mir221/222;Mir221/222 $^{d/f, f/-}$ (n = 3) and huTNFtg;TgCol6a1-Mir221/222;Mir221/222$^{-/-}$ mice (n = 7). Scale bars: 800 μm and 250 μm. Student's *t*-test statistical analysis was used, unpaired.

The online version of this article includes the following figure supplement(s) for figure 2:

**Figure supplement 1.** TgCol6a1-Mir221/222 mice target tissues and cells of mesenchymal origin.

**Figure supplement 2.** *Mir221/222* overexpression may affect bone physiology. Synovial fibroblasts (SFs) from Mir221/222$^{-/-}$ mice lack *Mir221/222* expression.

To uncover direct targets of *Mir221/222*, we cross-referenced downregulated genes in SFs from huTNFtg compared to WT and from huTNFtg;TgCol6a1-Mir221/222 compared to huTNFtg mice with their predicted targets using three different tools (DIANA-microT-CDS, Targetscan, and Pictar). Among these potential target candidates, we found *Cdkn1b*, *Cdkn1c* (being predicted by the three tools), and *Smarca1* (being predicted by DIANA-microT-CDS and Pictar) (**Figure 4B**). *Cdkn1b* and *Cdkn1c* are cell cycle inhibitors and have been previously validated as *Mir221/222* targets (**le Sage et al., 2007**; **Pineau et al., 2010**), while *Smarca1* (a component of the NURF complex involved in chromatin remodeling [**Clapier et al., 2017**]) is a new predicted target. To verify their regulation by *Mir221/222*, the expression levels of all targets were determined in SFs isolated from huTNFtg;TgCol6a1-Mir221/222 mice, as well as control mice. All *Cdkn1b*, *Cdkn1c,* and *Smarca1* were found to be downregulated in SFs from huTNFtg;TgCol6a1-Mir221/222 mice compared to huTNFtg mice both at the RNA and protein levels (**Figure 4C–E and H–L**). Together our data suggest that *Mir222/221* target specific cell cycle inhibitors (*Cdkn1b* and *Cdkn1c*) in order to induce proliferation of fibroblasts. Additionally, we also identified *Smarca1*, a novel *Mir221/222* target involved in chromatin dynamics.

ECM-related genes such as *Prg4* and *Itga3* were found to be downregulated in huTNFtg SFs. Interestingly, *Mir221/222* overexpression resulted in enhanced downregulation of these genes, probably through indirect mechanisms as they were not predicted to be direct targets (**Figure 4F and G**). Next, the observed downregulation of genes of the ECM led us to check the expression of additional genes related with a destructive behavior of SFs such as *Mmp3* and *Rankl* (*Tnfsf11*). However, no difference was observed in *Mmp3* and *Rankl* levels in arthritogenic SFs due to *Mir221/222* overexpression (**Figure 4—figure supplement 2A and B**).

We further analyzed if IL-6 levels in SFs were regulated by *Mir221/222* overexpression, as previously reported (**Yang and Yang, 2015**). We did not observe any changes neither in the IL-6 protein levels in supernatants of naïve or TNF stimulated SFs nor in *Il6* RNA levels in SFs from *Mir221/222* overexpressing mice (**Figure 4—figure supplement 2C and D**).

Additionally, *Tnf* has been reported to be a direct target of *Mir221* and *Mir222* in sepsis in macrophages (**Seeley et al., 2018**; **El Gazzar and McCall, 2010**). However, in the huTNFtg model of arthritis, this could not apply as the endogenous 3'UTR of the human *TNF* gene is replaced by the 3'UTR of the *HBB* gene (**Keffer et al., 1991**). Nevertheless, we checked if there are indirect signals that stem from *Mir221/222* overexpression in SFs and affect the already deregulated expression of human *TNF*. RNA levels of *TNF* were not altered in SFs from huTNFtg;TgCol6a1-Mir221/222 mice compared to huTNFtg (**Figure 4—figure supplement 2E**). Moreover, no difference in the levels of human TNF and no mouse TNF protein was detected in the sera or supernatants of cultured SFs from huTNFtg and huTNFtg;TgCol6a1-Mir221/222 mice (**Figure 4—figure supplement 2F and G**). Furthermore, in inflamed joints, SFs become activated and act as recruiters of leukocytes by increasing the expression of adhesion molecules. To examine whether *Mir221/222* overexpression rendered fibroblasts more activated, we measured the expression of activation markers, such as ICAM-1 and VCAM-1, but no differences were observed (**Figure 4—figure supplement 2H1**). It is therefore evident that *Mir221/222* overexpression shifts the gene expression of arthritogenic SFs to a more proliferative and less ECM-producing signature, without any evidence of switching to a more pro-inflammatory or activated state.

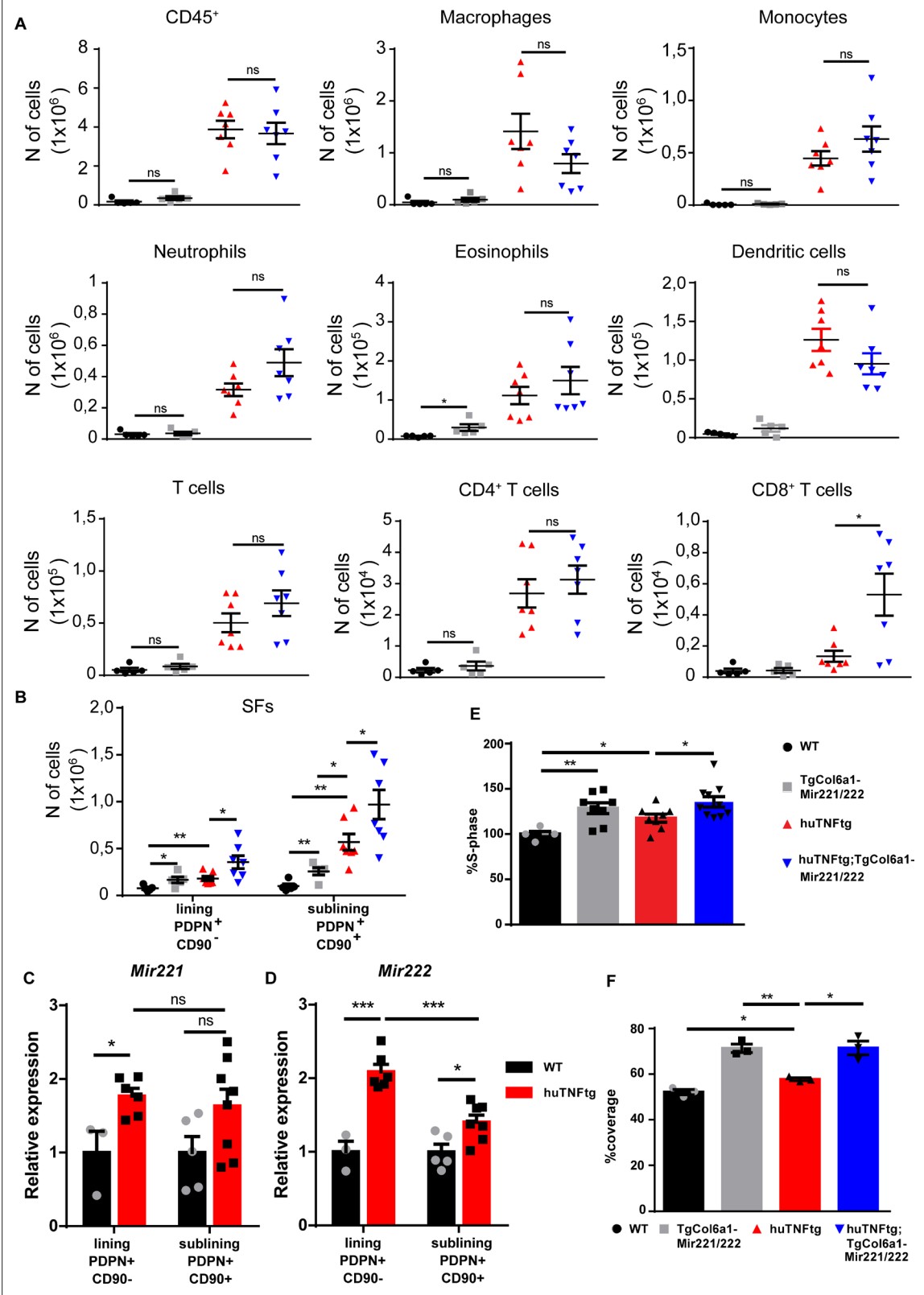

**Figure 3.** *Mir221/222* overexpression leads to fibroblast expansion. (**A**) Infiltration of CD45[+] cells, macrophages, monocytes, neutrophils, eosinophils, dendritic cells, CD4[+] T cells, and CD8[+] T cells in the ankle joints of 8-week-old WT (n = 5), TgColVI-Mir221/222 (n = 5), huTNFtg (n = 7), and huTNFtg;TgCol6a1-Mir221/222 mice (n = 7) quantified by FACS analysis (from two independent experiments). Student's *t*-test statistical analysis was used, unpaired. (**B**) Lining PDPN[+]CD90[-] and sublining PDPN[+]CD90[+] synovial fibroblast (SF) number quantification in the ankle joints of 8-week-old

*Figure 3 continued on next page*

*Figure 3 continued*

WT (n = 5), TgCol6a1-Mir221/222 (n = 5), huTNFtg (n = 7), and huTNFtg;TgCol6a1-Mir221/222 mice (n = 7) by FACS analysis (from two independent experiments). Student's *t*-test statistical analysis was used, unpaired. (**C, D**) *Mir221* and *222* levels in freshly sorted lining PDPN$^+$CD90$^-$ and sublining PDPN$^+$CD90$^+$ SFs in the ankle joints of 8-week-old WT and huTNFtg mice (n = 3–8). Student's *t*-test statistical analysis was used, unpaired. *Rnu6* was used as a housekeeping gene for normalization (**E**) % fraction of cultured SFs that are in the S-phase of the cell cycle from 8-week-old WT (n = 5), TgCol6a1-Mir221/222 (n = 8), huTNFtg (n = 8), and huTNFtg;TgCol6a1-Mir221/222 mice (n = 10) quantified by FACS analysis (from three independent experiments). Student's *t*-test statistical analysis was used, unpaired. (**F**) % area that was covered by cultured fibroblasts 24 hr after a wound was performed (n = 3). A representative experiment out of three is presented. Student's *t*-test statistical analysis was used, unpaired. All comparisons were performed using WT SFs as a reference sample. Data represent mean ± SEM. *p<0.05, **p<0.01, ***p<0.001, ****p<0,0001, ns = not significant.

The online version of this article includes the following figure supplement(s) for figure 3:

**Figure supplement 1.** FACS gating strategies.

**Figure supplement 2.** *Mir221/222* overexpression leads to increased effector CD8$^+$ T cell infiltration.

## Total deletion of *Mir221/222* ameliorates arthritis in huTNFtg mice

To assess the therapeutic potential of *Mir221/222* downregulation, we studied the impact of *Mir221/222* deletion in arthritis. Mir221/222$^{-/-}$ mice were crossed with huTNFtg mice to ablate *Mir221/222* during arthritis. huTNFtg;Mir221/222$^{-/-}$ mice presented with decreased synovial hyperplasia, cartilage destruction, and number of osteoclasts, as seen in histological analysis, compared to huTNFtg (*Figure 5A*). Additionally, *Mir221/222* deletion in arthritis did not alter the inflammatory influx (*Figure 5—figure supplement 1*), but rather led to the inhibition of fibroblast expansion of LL and SL in the joints (*Figure 5B*). Ex vivo, functional characterization of arthritic SFs in the absence of *Mir221/222* showed decreased proliferative capacity than SFs from huTNFtg mice (*Figure 5C*), confirming the control of cell proliferation by these two miRNAs. *Cdkn1b, Cdkn1c,* and *Smarca1* RNA and protein levels were partially restored (*Figure 5D–K*), verifying their regulation by *Mir221/222*, although alternative or feedback mechanisms might also exist preventing the complete restoration of the expression levels of these genes and complete disease protection of huTNFtg;Mir221/222$^{-/-}$ mice.

Finally, a luciferase assay was performed to validate that *Smarca1* is indeed a direct target of *Mir221/222* and its expression is not modulated by these two miRNAs via indirect mechanisms. As seen in *Figure 5L*, combined transfection of HEK293 cells with pre-*Mir221/222* and a plasmid carrying the *Mir221/222* binding site of *Smarca1* decreased luciferase activity contrary to transfections with scramble pre-*Mir* and/or a plasmid carrying a mutated *Mir221/222* binding site of *Smarca1*.

## *Mir221/222* gene activity marks the pathogenic clusters of expanding intermediate and lining compartment of SFs in arthritis

Our next step was to define *Mir221/222* gene activity scores at a single-cell level in arthritogenic SFs. Chromatin accessibility analysis of previously generated scATAC-seq data from ankle joint fibroblasts of WT and huTNFtg mice (*Armaka et al., 2022*) revealed the emergence and expansion of a pathogenic intermediate cluster and a destructive one in the lining (*Figure 6A*, *Figure 6—figure supplement 1A*), along with clusters found physiologically in WT mice. As can be seen, *Mir221/222* locus (*Mirc53*) is more accessible in the pathogenic intermediate and destructive lining clusters (*Figure 6B*, *Figure 6—figure supplement 1B and C*), marking their aberrant expansion. To gain insight into which regulatory elements may control *Mir221/222* expression in arthritogenic SFs, correlation analysis between chromatin accessibility and gene activity scores was performed in the regulatory space of *Mir221/222*. Inferred linkages between accessible chromatin (peaks) and *Mir221/222* genetic locus revealed transcription factor binding sites (TFBSs) of previously reported positive regulators such as *Nfe2l2, Junb, Rela, Relb* and *Bach1* (*Armaka et al., 2022*), to be enriched in the regulatory regions associated with *Mir221/222* locus (*Figure 6C*, *Figure 6—figure supplement 1D*). Overall, the increased accessibility of *Mir221/222* locus in the activated and expanding intermediate cluster, as well as in the destructive cluster of the lining layer in arthritis, corroborates the role of these two miRNAs in promoting proliferation and expansion of activated clusters of SFs in disease.

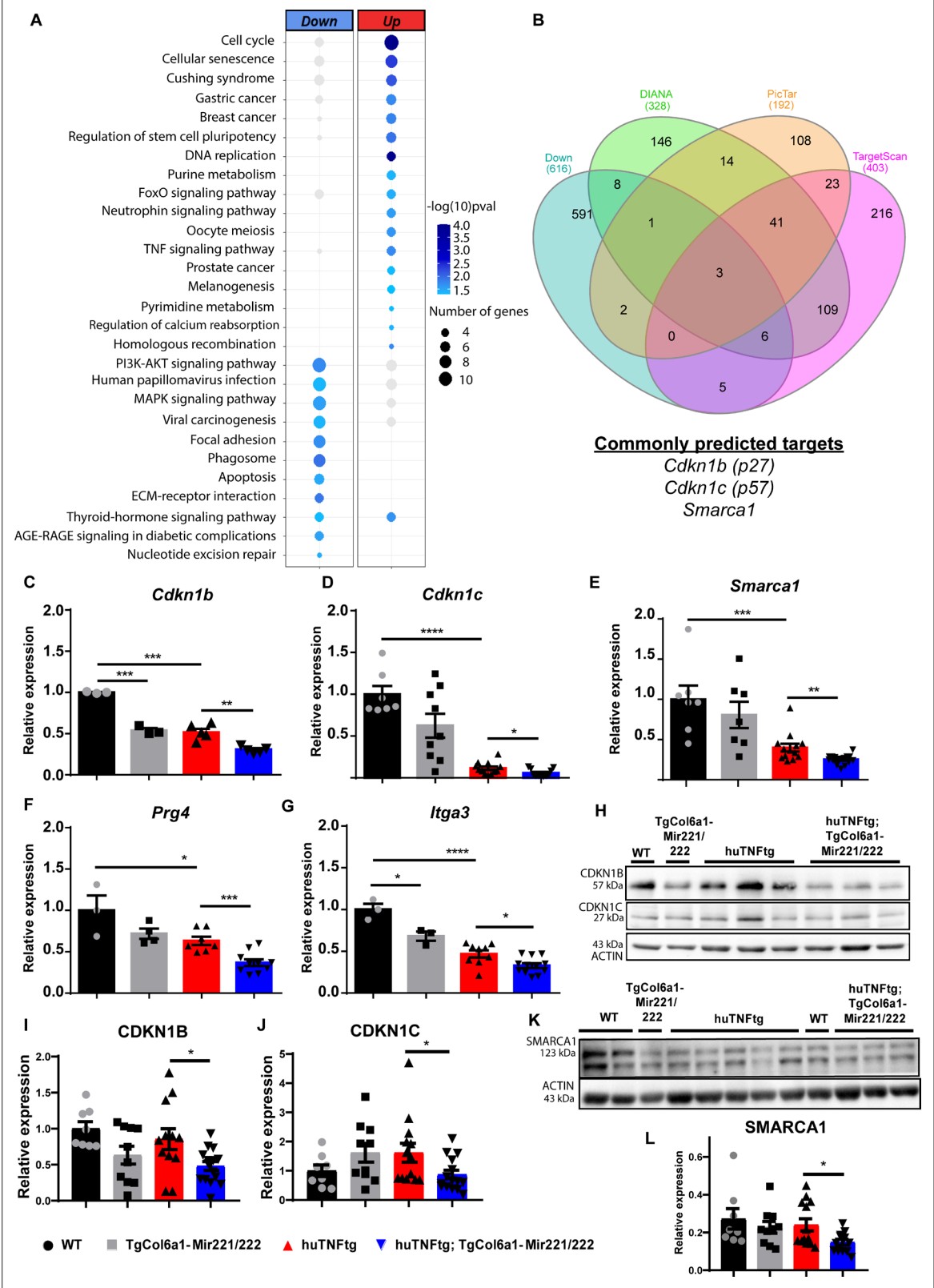

**Figure 4.** *Mir221/222* regulate cell cycle signaling and ECM-related pathways in arthritis. (**A**) Bubble plot of enriched KEGG pathways in the up- (red) and downregulated genes (blue) originated from the comparison of huTNFtg;TgCol6a1-Mir221/222 and huTNFtg bulk RNA-seq profiles. The color of the bubble signifies the statistical significance, while the size denotes the number of up-/downregulated genes found in the enriched term. (**B**) Venn diagram showing the overlap of genes predicted as *Mir221/222* targets from DIANA, PicTar, and TargetScan and downregulated genes from huTNFtg

*Figure 4 continued on next page*

_Figure 4 continued_

compared to WT and huTNFtg;TgCol6a1-Mir221/222 compared to huTNFtg synovial fibroblasts (SFs). (**C–G**) Expression analysis as defined by qRT-PCR of _Cdkn1b, Cdkn1c, Smarca1, Prg4,_ and _Itga3_ in cultured SFs from 8-week-old WT, TgCol6a1-Mir221/222, huTNFtg, and huTNFtg;TgCol6a1-Mir221/222 mice (n = 3–16). WT SFs were used as a reference. Data from 2 to 4 independent experiments. Student's _t_-test statistical analysis was used, unpaired. In all experiments, _B2m_ was used as a housekeeping gene for normalization. (**H**) Representative western blots depicting protein expression analysis of CDKN1B and CDKN1C in cultured SFs from 8-week-old WT, TgCol6a1-Mir221/222, huTNFtg, and huTNFtg;TgCol6a1-Mir221/222 mice. (**I, J**) Quantification of protein expression analysis as defined by western blots of CDKN1B and CDKN1C in cultured SFs from 8-week-old WT, TgCol6a1-Mir221/222, huTNFtg, and huTNFtg;TgCol6a1-Mir221/222 mice (n = 8–15). WT SFs were used as a reference. Data from four independent experiments. Student's _t_-test statistical analysis was used, unpaired. In all experiments, ACTIN was used as a housekeeping gene for normalization. (**K**) Representative western blot depicting protein expression analysis of SMARCA1 in cultured SFs from 8-week-old WT, TgCol6a1-Mir221/222, huTNFtg, and huTNFtg;TgCol6a1-Mir221/222 mice. (**L**) Quantification of protein expression analysis as defined by western blots of SMARCA1 in cultured SFs from 8-week-old WT, TgCol6a1-Mir221/222, huTNFtg, and huTNFtg;TgCol6a1-Mir221/222 mice (n = 8–13). WT SFs were used as a reference. Data from four independent experiments. Student's _t_-test statistical analysis was used, unpaired. In all experiments, ACTIN was used as a housekeeping gene for normalization. Data represent mean ± SEM. *p<0.05, **p<0.01, ***p<0.001, ****p<0,0001, ns = not significant.

The online version of this article includes the following source data and figure supplement(s) for figure 4:

**Source data 1.** Differentially expressed genes as detected in bulk RNA sequencing originating from the comparison between different genotypes.

**Source data 2.** List depicting downregulated genes stemming from the comparisons of huTNFtg compared to WT and huTNFtg;TgCol6a1-Mir221/222 compared to huTNFtg SFs along with _Mir221/222_ predicted targets using three different tools (DIANA-microT-CDS, Targetscan, and Pictar).

**Source data 3.** Over-represented and down-represented KEGG pathways in the RNA expression profile of huTNFtg;TgCol6a1-Mir221/222 SFs compared to the huTNFtg.

**Source data 4.** Uncropped blot of CDKN1B, CDKN1C, and ACTIN in cultured SFs from 8-week-old WT, TgCol6a1-Mir221/222, huTNFtg, and huTNFtg;TgCol6a1-Mir221/222 mice.

**Source data 5.** Uncropped blot of CDKN1B, CDKN1C, and ACTIN in cultured SFs from 8-week-old WT, TgCol6a1-Mir221/222, huTNFtg, and huTNFtg;TgCol6a1-Mir221/222 mice.

**Source data 6.** Uncropped blot of SMARCA1 and ACTIN in cultured SFs from 8-week-old WT, TgCol6a1-Mir221/222, huTNFtg, and huTNFtg;TgCol6a1-Mir221/222 mice.

**Figure supplement 1.** Comparisons between samples used for bulk RNA sequencing.

**Figure supplement 2.** _Mir221/222_ do not regulate inflammatory signaling in synovial fibroblasts (SFs).

## Discussion

miRNAs are important fine-tuners of gene expression and often exhibit a specific tissue or developmental expression pattern (_Inui et al., 2010_). Deregulation in miRNA expression levels has been linked with diseases such as cancer, viral infection, and inflammation (_O'Connell et al., 2012_).

In RA, alteration in expression of several miRNAs has been reported in different cell types in the synovium or in circulating peripheral blood mononuclear cells (PBMCs). _Mir146a_ and _Mir155_ are among the most highly studied miRNAs and both are upregulated in response to inflammatory signals and implicated in immune responses (_Nakasa et al., 2011_; _Saferding et al., 2017_; _Blüml et al., 2011_; _Kurowska-Stolarska et al., 2011_); however, in the vast majority of the studies, no tissue specificity for the function of miRNAs in arthritis was considered.

In this study, we try to shed light on the in vivo role of the _Mir221/222_ family of miRNAs in arthritis. In the past, IL-1β had been reported to mediate downstream signaling of TNF/TNFR1 axis (_Brennan et al., 1989_; _Probert et al., 1995_; _Zwerina et al., 2007_). In the present study, we provide evidence that in the arthritic microenvironment of the joints, TNFR1 signaling downstream of TNF and independently of IL-1β can induce _Mir221/222_ levels in SFs apart from LPS and could suggest that increased TNF and IL-1β levels can serve as signals that lead to _Mir221/222_ upregulation specifically in RA SFs.

In RA, proliferation of fibroblasts consists of one of the hallmarks of the aggressive behavior of the pathogenic mesenchyme, and it is mediated by survival and anti-apoptotic signals in SFs (_Pap et al., 2000_). Although the signals and epigenetic alterations that lead to this aberrant proliferation of SF subsets in RA remain under investigation, in this study we uncovered a novel role for _Mir221/222_ in the mesenchyma, where increased levels of _Mir221/222_ rendered fibroblasts more proliferative and migratory, even in the absence of an inflammatory trigger. Moreover, a causal role for _Mir221/222_ involvement in pannus hyperplasia was shown by the total ablation of these two miRNAs in the arthritic mice that led to decreased disease manifestations due to attenuated proliferation of SFs and decreased numbers of fibroblasts comprising the lining and sublining layers. Also, considering that

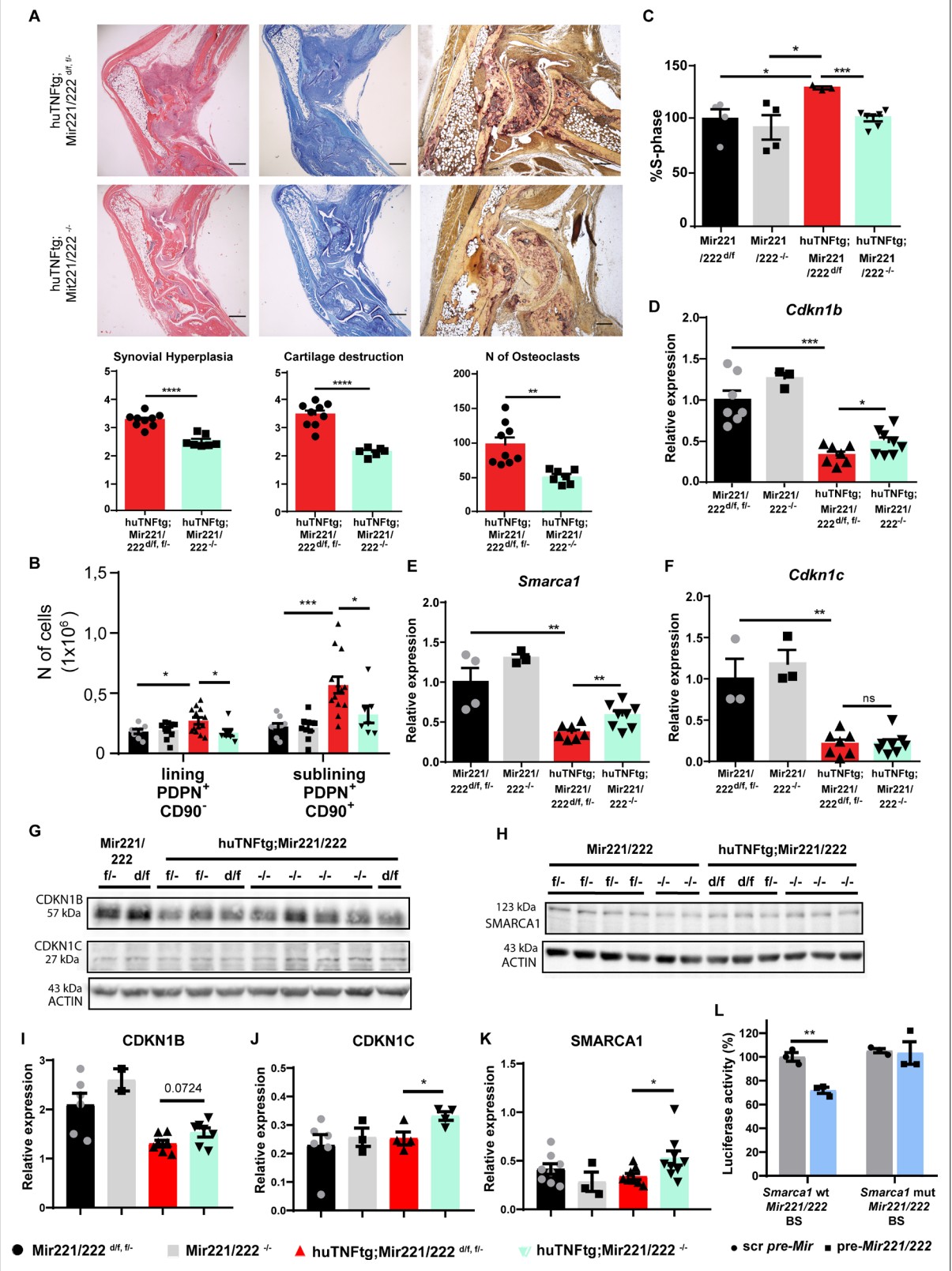

**Figure 5.** Deletion of *Mir221/222* ameliorates arthritis in huTNFtg mice. (**A**) Representative histological images of H&E, TB, and TRAP stained ankle joint sections and histological score of synovial hyperplasia, cartilage destruction, and osteoclast numbers from 9-week-old huTNFtg;Mir221/222 d/f, f/- (n = 9) and huTNFtg;Mir221/222 -/- mice (n = 6–7). Student's t-test statistical analysis was used, unpaired. (**B**) Lining PDPN+CD90- and sublining PDPN+CD90+ synovial fibroblast (SF) number quantification in the ankle joints of 9-week-old Mir221/222 f/-, d/f (n = 8–9), Mir221/222 -/- (n = 10), huTNFtg;Mir221/222 f/-,

*Figure 5 continued on next page*

*Figure 5 continued*

[d/f] (n = 13) and huTNFtg;Mir221/222 [-/-] mice (n = 7) by FACS analysis (from three independent experiments). Student's *t*-test statistical analysis was used, unpaired. (C) % fraction of cultured SFs that are in the S-phase of the cell cycle from 9-week-old Mir221/222 [d/f] (n = 4), Mir221/222 [-/-] (n = 4), huTNFtg;Mir221/222 [d/f] (n = 3), and huTNFtg;Mir221/222 [-/-] (n = 7) (from two independent experiments). Mir221/222 [d/f] SFs were used for normalization. Student's *t*-test statistical analysis was used, unpaired. (D–F) Expression analysis as defined by qRT-PCR of *Cdkn1b*, *Cdkn1c*, and *Smarca1* in cultured SFs from 9-week-old Mir221/222 [f/-, d/f], Mir221/222 [-/-], huTNFtg;Mir221/222 [d/f] and huTNFtg;Mir221/222 [-/-] (n = 3–8, from two independent experiments). Expression of Mir221/222 [d/f, f/-] SFs was used for normalization. Student's *t*-test statistical analysis was used, unpaired. In all experiments, *B2m* was used as a housekeeping gene for normalization. (G, H) Representative western blots depicting protein expression analysis of CDKN1B, CDKN1C, and SMARCA1 in cultured SFs from 9-week-old Mir221/222 [f/-, d/f], Mir221/222 [-/-], huTNFtg;Mir221/222 [d/f], and huTNFtg;Mir221/222 [-/-] mice. (I–K) Quantification of protein expression analysis as defined by western blots of CDKN1B, CDKN1C, and SMARCA1 in cultured SFs from 9-week-old Mir221/222 [f/-, d/f], Mir221/222 [-/-], huTNFtg;Mir221/222 [d/f], and huTNFtg;Mir221/222 [-/-] mice (n = 2–9). WT SFs were used as a reference. Data from 2 to 3 independent experiments. Student's *t*-test statistical analysis was used, unpaired. In all experiments, ACTIN was used as a housekeeping gene for normalization. (L) Luciferase activity in HEK293 cells transfected with constructs containing the wt *Mir221/222* binding site (BS) of *Smarca1* or the mutated one and co-transfected with either the control scramble pre-*Mir* or the pre-*Mir221/222* for 72 hr. Scramble pre-*Mir* control co-transfected with the wt *Smarca1 Mir221/222* BS served as a reference. Data from three independent experiments. Student's *t*-test statistical analysis was used, unpaired. Scale bars: 600 μm and 300 μm. Data represent mean ± SEM. *p<0.05, **p<0.01, ***p<0.001, ****p<0,0001, ns = not significant.

The online version of this article includes the following source data and figure supplement(s) for figure 5:

**Source data 1.** Uncropped blot of CDKN1B, CDKN1C, and ACTIN in cultured SFs from 9-week-old Mir221/222 f/-, d/f, huTNFtg;Mir221/222 d/f, and huTNFtg;Mir221/222-/- mice.

**Source data 2.** Uncropped blot of CDKN1B, CDKN1C, and ACTIN in cultured SFs from 9-week-old Mir221/222 f/-, d/f, huTNFtg;Mir221/222 d/f, and huTNFtg;Mir221/222-/- mice.

**Source data 3.** Uncropped blot of CDKN1B, CDKN1C, and ACTIN in cultured SFs from 9-week-old Mir221/222 f/-, d/f, huTNFtg;Mir221/222 d/f, and huTNFtg;Mir221/222-/- mice.

**Source data 4.** Uncropped blot of SMARCA1 and ACTIN in cultured SFs from 9-week-old Mir221/222 f/-, d/f, Mir221/222-/-, huTNFtg;Mir221/222 d/f, and huTNFtg;Mir221/222-/- mice.

**Figure supplement 1.** *Mir221/222* deletion in experimental arthritis does not regulate inflammatory influx in the joints.

mesenchymal overexpression of *Mir221/222* at naïve conditions (WT background) led to a tendency towards worse bone integrity and increased fibroblast populations suggests that *Mir221/222* upregulation and control of target gene expression could have dramatic effects in a specific milieu under local unresolved inflammation.

Analysis of the transcriptional profile of SFs from huTNFtg;TgCol6a1-Mir221/222 mice compared to huTNFtg mice uncovered deregulation in pathways linked to cell cycle and ECM regulation. Cell cycle inhibitors *Cdkn1b* and *Cdkn1c*, known *Mir221/222* targets, were both identified in our data and were verified by ex vivo experiments in SFs. Interestingly, deregulation in ECM components was observed due to *Mir221/222* overexpression. ECM plays a crucial role in the joints and apart from providing support, it also promotes cell communication and motility. Synovial fluid containing hyaluronic acid and lubricin (PRG4) protects from friction and lubricates the cavity area (*Buckley et al., 2021*). Interestingly, during *Mir221/222* overexpression in arthritis, the expression of ECM-related molecules in SFs was found altered and, more specifically, *Prg4* and the integrin component *Itga3* were downregulated. *Prg4* and *Itga3* were not predicted as direct targets of *Mir221/222*, so indirect mechanisms probably mediate this effect. *Itga3* expression has been found to be downregulated in PBMCs from OA (osteoarthritis) patients (*Zhang et al., 2019*), and a previous study has reported that the absence of *Prg4* from the synovium leads to OA and aberrant fibroblast expansion and proliferation (*Rhee et al., 2005*). Moreover, a cardioprotective and antifibrotic role of *Mir221/222* was observed in cardiac fibroblasts by the downregulation of different types of collagens (*Zhou et al., 2019*). So, *Mir221/222* could contribute to the catabolic profile of SFs in arthritis by affecting expression of ECM components. Furthermore, the mesenchymal overexpression of *Mir221/222* did not render SFs more inflammatory or activated per se as it was supported by ex vivo studies in the past (*Yang and Yang, 2015*). In contrast, our in vivo data suggest that these two miRNAs regulate pro-proliferative signals leading to aberrant expansion of pathogenic SFs that in the end renders the arthritic milieu more inflammatory.

Additionally, *Smarca1* was identified as a new direct target of *Mir221/222* based on the transcriptional profile of SFs, bioinformatic analysis of predicted targets, and luciferase assay. SMARCA1 participates in the NURF chromatin remodeling complex that can limit or promote chromatin accessibility, and thus block or activate gene expression correspondingly (*Clapier et al., 2017*). NURF complex has

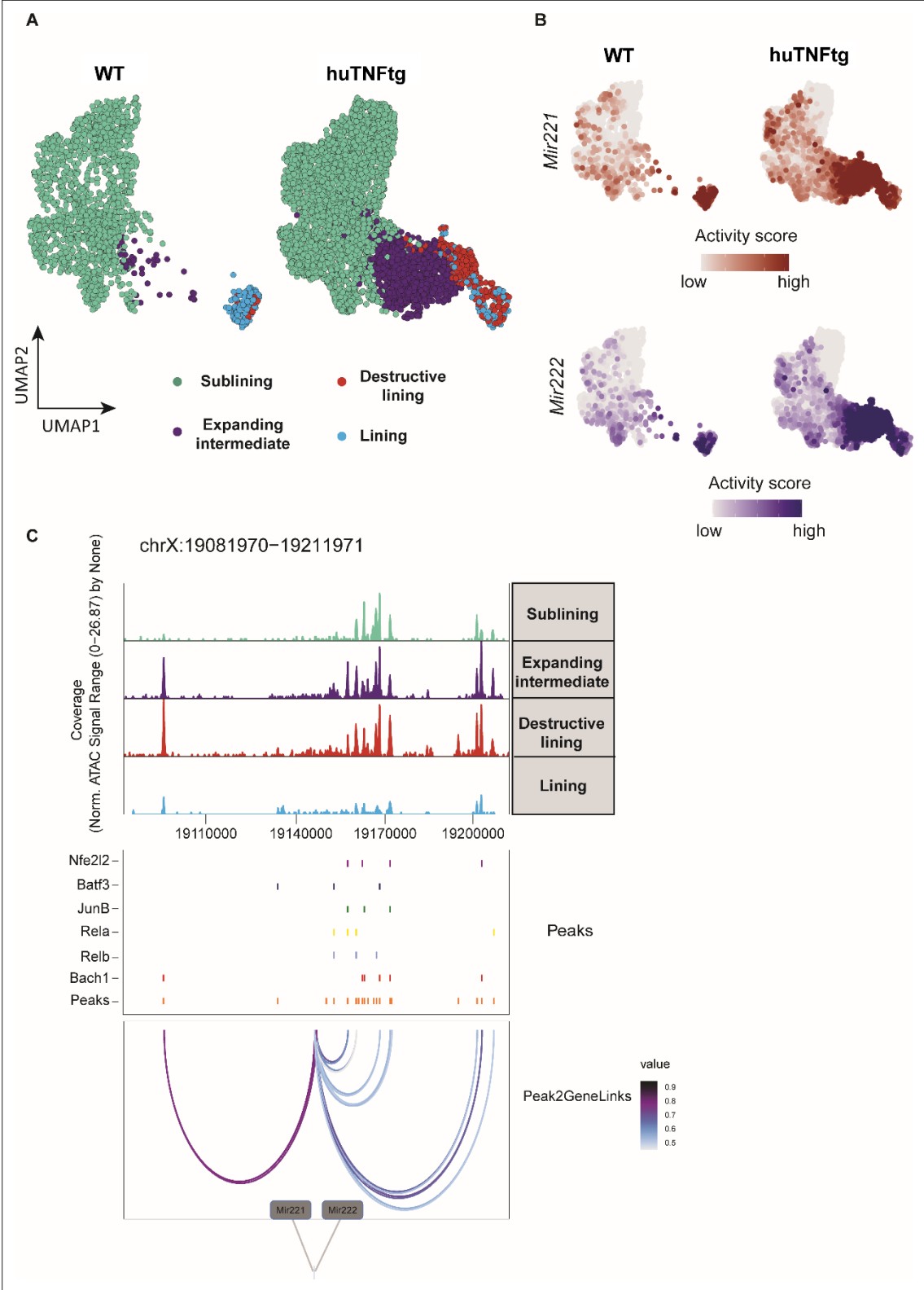

**Figure 6.** *Mir221/222* gene activity is increased in the pathogenic subclusters of the expanding intermediate and lining compartment. (**A**) UMAP projection of 6046 synovial fibroblast nuclei obtained by scATAC-seq from WT and huTNFtg samples. Cells are grouped in four categories: sublining (light green), expanding intermediate (purple), lining (light blue), and destructive lining (red). (**B**) Feature plots, in UMAP space, depicting gene activity scores of *Mir221* (red) and *Mir222* (blue) in WT and huTNFtg samples. (**C**) Genome accessibility track visualization of the extended regulatory space of *Mir221* and *Mir222* (chrX:19,081,970–19,211,971), with TF binding site and peak-to-gene linkage information, in huTNFtg samples. Upper: the genome

*Figure 6 continued on next page*

*Figure 6 continued*

track shows increased accessibility in expanding intermediate and destructive lining clusters. Middle: all reproducible peaks are shown, coupled with annotated CISBP binding information for *Bach1*, *Rela*, *Relb*, *Nfe2l2*, *Batf3*, and *Junb* TFs. Lower: putative regulatory linkages between *Mir221-Mir222* genes and reproducible peaks are illustrated. Links between genes and peaks are colored by correlation (Pearson coefficient) of peak accessibility and gene activity scores.

The online version of this article includes the following figure supplement(s) for figure 6:

**Figure supplement 1.** *Mir221/222* gene activity in synovial fibroblast (SF) clusters in normal and arthritic state.

been shown to promote cell proliferation, but a role for this complex in RA is missing (***Ding et al., 2019***). Future studies will reveal a specific role for SMARCA1 in shaping gene expression in arthritis. Recently, a study reported a role for *Mir222* in targeting *Brg1*, an SWI/SNF chromatin remodeling component, leading to repression of inflammatory cytokine expression in sepsis (***Seeley et al., 2018***).

Finally, analysis of chromatin accessibility data at a single-cell level revealed increased chromatin accessibility in *Mir221/222* locus in the activated and expanding intermediate cluster and in the destructive cluster of the lining. This analysis, coupled with transcription factor binding site linkage, revealed transcription factors such as *Junb*, *Rela*, *Relb* (***Galardi et al., 2011***), *Nfe2l2*, and *Bach1*, which may play a role in positively regulating the activity of these two miRNAs in arthritis, leading to the expansion of pathogenic and destructive clusters of the intermediate and lining compartments, respectively. Notably, there is a recent study uncovering a role for NRF2 and BACH1 in promoting NSCLC tumor metastasis (***Lignitto et al., 2019***). Future experiments may thus elucidate the exact role of these transcription factors in the aggressive, autonomous, and migratory character of RA SFs.

We observed *Mir221* and *Mir222* induction to reach peak levels in late disease stages, and this agrees with a previous study that correlates their levels with disease activity in RA patients (***Abo ElAtta et al., 2019***). Moreover, their overexpression is sufficient to worsen the severity of the disease. Thus, these miRNAs could serve as biomarkers to predict disease progression in humans.

Additionally, therapeutic targeting of *Mir221/222* could be beneficial as total genetic deletion of these two miRNAs protected to a certain degree from arthritis progression and the expression of their confirmed targets was partially restored. This partial protection underlines the fact that most probably there are additional mechanisms assuring the suppression of these targets and should be addressed. It is known that miRNA function is often redundant, which means that if an miRNA is missing, other miRNAs can regulate its targets. Further studies are therefore needed in order to identify additional miRNAs that target the same genes and assess the potential of their combined downregulation in the development of new treatments. Nonetheless, it is very interesting that during arthritis there are redundant and complementary pathways ensuring expansion of the pathogenic stroma subsets. I t is of great interest to dissect which pro-survival and pro-proliferative pathways lead to expansion of pathogenic fibroblast subpopulations that could constitute novel targets for subpopulation-specific therapeutic intervention.

Collectively, we report that *Mir221* and *Mir222* lead to enhanced disease progression in a mouse model of inflammatory polyarthritis due to increased pathogenic SF proliferation under inflammatory conditions, mainly through downregulation of cell cycle inhibitors. Finally, *Mir221/222* gene activity is increased in the pathogenic and expanding intermediate and destructive lining clusters. Thus, mesenchymal targeting of *Mir221/222* could potentially serve as a therapeutic tool for treating RA and their expression could be used as biomarkers for predicting disease outcomes.

## Materials and methods

### Mice

Human TNF transgenic (huTNFtg), Tnfr1$^{-/-}$(***Rothe et al., 1993***), and Deleter-Cre (***Schwenk et al., 1995***) mice have been previously described. Mir221/222 $^{f/f}$ were purchased from the European Mouse Mutant Archive (EMMA) provided by Helmholtz Zentrum Muenchen (EMMA ID: EM 05507). Deleter-Cre were used to generate Mir221/222$^{-/-}$ mice. TgCol6a1-Mir221/222 mice were generated in Kollias Lab, BSRC Al. Fleming Institute. Briefly, around 1 kbp containing the *Mir221/222* (*Mirc53*) locus was inserted into the BpII site of intron 2 of the *HGH* minigene in a pBluescript SK (+) vector. Then, the *Col6α1* promoter was inserted upstream of the previous fragment in the SaII–BamHI (blunt) site of the pBluescript vector. The Col6a1::Mir221/222 transgene (around 11 kbp) was excised as a SaII–NotI

fragment for pronuclear microinjections and generation of TgCol6a1-Mir221/222 mice. Genetic analysis (by Southern blot) of the transgenic TgCol6a1-Mir221/222 mice showed that the transgene is inserted in a head-to-tail direction and in approximately 20 copies. All mice (both sexes) used in the study were analyzed compared to their respective controls that were cohoused littermates. Mice were euthanized by $CO_2$ exposure.

All mice were maintained in a C57BL/6J or CBA;C57BL/6J genetic background. Mice were maintained under specific pathogen-free conditions in conventional, temperature-controlled, air-conditioned animal house facilities of BSRC Al. Fleming with 12 hr light/12 hr dark cycle and received food and water ad libitum.

## Histology

Formalin-fixed, EDTA-decalcified, paraffin-embedded mouse joint tissue specimens were sectioned and stained with hematoxylin-eosin (H&E), Toluidine Blue, or Tartrate-Resistance Acid Phosphatase (TRAP) Kit (Sigma-Aldrich). H&E and TB were semi-quantitatively blindly evaluated for the following parameters: synovial inflammation/hyperplasia (scale of 0–5) and cartilage erosion (scale of 0–5) based on an adjusted, previously described method (*Mould et al., 2003*). TRAP staining of joint sections was performed to measure number of osteoclasts using ImageJ software. Images were acquired with a Nikon microscope, equipped with a QImaging digital camera.

## Microcomputed tomography

Microcomputed tomography (microCT) of excised joints was carried out using a SkyScan 1172 CT scanner (Bruker, Aartselaar, Belgium) at BSRC Al. Fleming Institute following the general guidelines used for the assessment of bone microarchitecture in rodents using microCT (*Bouxsein et al., 2010*). Briefly, scanning was conducted at 50 kV, 100 mA using a 0.5 mm aluminum filter, at a resolution of 5 mm/pixel. Reconstruction of sections was achieved using the NRECON software (Bruker) with beam hardening correction set to 40%. The analysis was performed on a spherical volume of interest (diameter 0.54 mm) within 73 slides of the trabecular region of calcaneus. Morphometric quantification of trabecular bone indices such as trabecular bone volume fraction (BV/TV%), bone surface density (BS/TV%), trabecular number (Tb. N; 1/mm), and trabecular separation (Tb. Sp; mm) were performed using the CT analyzer program (Bruker). Additionally, 3D images of the scanned area of the samples were generated using the CTvox software (Bruker).

## Cell culture

Primary mouse SFs were isolated from the ankle joints of mice (*Armaka et al., 2009*) with the indicated genotypes and were depleted from $CD45^+$ cells using Biotinylated anti-mouse CD45 antibody (BioLegend) and Dynabeads Biotin Binder (Invitrogen) according to the manufacturer's instructions and cultured for 3–4 passages for subsequent experiments.

SFs were seeded at a concentration of $8 \times 10^5$ on a 60 mm plate, starved overnight, stimulated the next day with either huTNF (10 ng/ml), LPS (100 ng/ml), mIL-1β (10 ng/ml), IFN-γ (10 ng/ml), or PolyIC (20 μg/ml) and collected at the indicated time points.

In experiments using the IL-1 inhibitor (Anakinra), pretreatment of cells with Anakinra at a concentration of 10 μg/ml was performed for 4 hr before stimulation with mIL-1β (10 ng/ml) or huTNF (10 ng/ml) and subsequent analysis.

## FACS analysis

FACS analysis and sorting of SF subpopulations were performed by removing ankle joints, cutting them into pieces and being digested using 1000 U/ml Collagenase IV (Sigma-Aldrich) in DMEM for 60 min at 37°C. The cell suspension was centrifuged, resuspended in FACS buffer (PBS with 0.5% FBS, 0.05% sodium azide, and 2 mM EDTA), and cells were counted. The anti-Fc Receptor (anti-CD16/32) antibody (BioLegend 101302) was used to prevent nonspecific binding. For stainings, 1–2 million cells were incubated with the following antibodies: PE-conjugated anti-CD11b (BD Biosciences, 557397), APC/Cy7- or A700-conjugated anti-CD45 (BioLegend 103116, 103128), APC-conjugated anti-MHCII (eBioscience, 17-5320-82), PE/Dazzle594-conjugated anti-CD64 (BioLegend, 139320), APC/Cy7-conjugated anti-CD24 (BioLegend, 101840), FITC-conjugated anti-Ly6C (BD Biosciences, 553104), PE/Cy7-conjugated anti-CD11c (BioLegend, 117318), PE-conjugated anti-B220

(BD Biosciences, 553089), PE/Cy7-conjugated anti-CD3 (eBioscience, 25-0031-82), A700-conjugated anti-CD4 (BioLegend, 100536), APC-conjugated anti-CD8 (BioLegend, 100711), PerCP-Cy5.5-conjugated anti-CD62L (eBioscience, 45-0621-80), APC-eFluor780 anti-CD44 (eBioscience, 47-0441-80), A488-conjugated anti-CD90.2 (BioLegend, 105316), PE-conjugated anti-CD31 (BD Biosciences, 553373), PE/Cy7-conjugated anti-PDPN (BioLegend, 127412), biotinylated anti-Ly6G (eBioscience, 13-5931-75), streptavidin-conjugated PE/Cy5 (BD Pharmingen, 554062), PE-conjugated anti-ICAM-1 (BD Pharmingen, 553253), and A647-conjugated anti-VCAM-1 (BioLegend, 105712). Zombie Green (Sigma 423112) or NIR (Sigma 77184) and DAPI (Invitrogen D1306) were used for live/dead exclusion. Analysis was performed using a FACS Canto II Flow cytometer (BD Biosciences) and FACS Diva (BD Biosciences) or FlowJo software (FlowJo, LLC), and cell sorting was performed using a FACS Aria III Cell Sorter (BD Biosciences).

## Western blot
Samples were collected in RIPA buffer (1% NP-40, 0.1% sodium dodecyl sulfate [SDS], 150 mM NaCl, 50 mM Tris–HCl, pH 7.4, 1 mM EDTA supplemented with protease [Roche] and phosphatase inhibitors [Sigma-Aldrich]), separated by SDS-PAGE (8 and 12.5%), transferred to nitrocellulose membranes (Millipore), and probed overnight with the following Abs: ACTIN (sc-1615, Santa Cruz Biotechnology, 1:2000), CDKN1B (3688, Cell Signaling, 1:1000), CDKN1C (ab-4058, Abcam, 1/800), and SMARCA1 (ab-172483, Abcam, 1/1000). Relative quantification was performed using ImageJ software analysis tool.

## Cell cycle analysis
Cell cycle was detected by flow cytometry. Briefly, $4 \times 10^5$ cells were seeded onto 60 mm plates and incubated for 24 hr. The cells were then harvested and washed with PBS. Next, the pellet was resuspended and fixed in 70% prechilled ethanol for 30 min at 4°C. The cells were washed again with PBS followed by treatment with RNase for 30 min at 37°C and addition of 200 µl staining solution with propidium iodide into the pellet. The final mixture was analyzed by flow cytometry.

## ELISA
Supernatants were collected at 0, 24, and 48 hr, and IL-6 quantification was performed using the mouse IL-6 Duo-Set ELISA kit, according to the manufacturer's instructions (R&D Systems).

## Wound-healing assay
SFs were plated to form a monolayer on wells of a 24-well plate, serum-starved overnight, and wounded with 200 µl pipette tips the next day. The culture dishes were washed three times with 1×PBS to remove detached cells, and the remaining cells were grown in DMEM containing 10% FBS. After 24 hr of incubation, wound-healing potential was quantified by counting the surface that cells had migrated, proliferated, and covered using ImageJ software analysis tool.

## RNA isolation and qRT-PCR
RNA was isolated from SFs, IMCs, spleenocytes, and epithelial cells using the RNeasy or the miRNeasy mini kit (QIAGEN) according to the manufacturer's instructions. Isolated RNA was subsequently used either for 3′RNA-seq sequencing and analysis or for construction of cDNA using the MMLV Reverse Transcriptase (Promega) or the TaqMan MicroRNA Reverse Transcription kit (Applied Biosystems). The cDNA was subsequently used for qRT-PCR using the Platinum SYBR-Green qPCR SuperMix (Invitrogen) or for detection of miRNA levels the TaqMan gene expression mastermix (Applied Biosystems) and the relative TaqMan microRNA probes. The CFX96 Touch Real-Time PCR Detection System (Bio-Rad) was used. Quantification was performed with the DDCt method.

## 3′ RNA-seq sequencing
The quantity and quality of RNA samples were analyzed using Agilent RNA 6000 Nano kit with the bioanalyzer from Agilent. RNA samples with RNA Integrity Number (RIN) > 7 were used for library construction using the 3′ mRNA-Seq Library Prep Kit Protocol for Ion Torrent (QuantSeq-LEXOGEN) according to the manufacturer's instructions. DNA High Sensitivity Kit in the bioanalyzer was used to assess the quantity and quality of libraries according to the manufacturer's instructions (Agilent).

Libraries were then pooled and templated using the Ion PI IC 200 Kit (Thermo Fisher Scientific) on an Ion Proton Chef Instrument or Ion One Touch System. Sequencing was performed using the Ion PI Sequencing 200 V3 Kit and Ion Proton PI V2 chips (Thermo Fisher Scientific) on an Ion Proton System according to the manufacturer's instructions.

## Luciferase reporter gene assays

The wt *Mir221/222* binding site (BS) in the *Smarca1* 3'UTR (5' AAACTAGCGGCCGCACAATGCTT TCTACCTGAAATGTGTAGCTT 3') and a mutated *Smarca1 Mir221/222* binding site (5' AAACTAGC GGCCGCACTTTGGATTGTAGGTCTAATGACATCGAT 3') were cloned in the pmirGLO Dual Luciferase miRNA target expression vector (Promega) downstream of the luciferase gene. Then, HEK293 cells were seeded in a 96-well plate at a density of $10^4$ cells/well 24 hr prior transfection. The next day, cells were transiently co-transfected using Lipofectamine 3000 with 100 ng cloned pmirGLO plasmid and 50 nM pre-*Mir221* and pre-*Mir222* mimic or control scramble pre-*Mir* for 72 hr. Samples were lysed using the Dual-Glo Luciferase Assay System (Promega) and luciferase activity was measured according to the manufacturer's instructions. Firefly was divided by Renilla activity and normalized to the scramble pre-*Mir* control co-transfected with the wt *Smarca1 Mir221/*222 BS that served as a reference.

## 3' RNA-seq bioinformatics analysis

Quality of the FASTQ files (obtained after Ion Proton sequencing) was assessed by FastQC, following the software recommendations. Alignment of sequencing reads to the mouse reference genome was performed using the software HISAT2 (version 2.1.0) with the mm10 version of the reference genome. The raw bam files were summarized to read counts table using FeatureCounts (version 1.6.0) and the gene annotation file mus_musculus.grcm38.92.gtf from Ensembl database. The resulting gene counts table was subjected to differential expression analysis utilizing the R package DESeq2. Differentially expressed genes were identified after setting the following thresholds: $p$-value$<0.05$ and $|log2FC|>1$. Functional enrichment analysis of the up- and downregulated genes for the contrasts of interest was performed with the web version of the tool enrichR. Enriched KEGG pathways (KEGG Mouse 2019) were selected by setting the following thresholds: $p$-value$<0.05$ and gene count $> 2$.

## scATAC-seq bioinformatics analysis

Analysis of scATAC-seq datasets was conducted by using the cellranger (10X Genomics) and ArchR suite as previously described (*Armaka et al., 2022*; *Granja et al., 2021*). Briefly, BCL files were converted to FASTQ files, aligned to UCSC mm10 reference genome, and WT and huTNFtg samples were aggregated and counted using 500 bp bins (tiles). Latent Semantic Indexing, graph-based clustering (louvain), and UMAP dimensionality reduction were applied accordingly. Gene activity scores (predictions of the level of expression of each gene) were computed as described in ArchR. Gene scores were scaled to 10,000 counts and log-normalized. To assign scATAC-seq cell-type identity, gene activity scores and scRNA-seq gene expression (*Armaka et al., 2022*) were aligned directly using a two-stage canonical correlation analysis. Non-fibroblast cells were excluded to result in 6046 SFs cells, which were reanalyzed as described above. The integration process between scATAC-seq and scRNA-seq SFs labeled the scATAC-seq cells according to four SF subpopulations (homeostatic, intermediate, lining, and destructive lining), which were visualized in a UMAP 2D space. Peak calling was applied in two pseudo-bulk replicates across all SFs (*Granja et al., 2021*), and then merged using an iterative overlap peak-merging (*Corces et al., 2018*). To identify enriched motifs in single-cell resolution, chromVar was used (*Schep et al., 2017*). Positive TF regulators were identified by correlating TF motif accessibility with integrated TF gene expression (Pearson correlation coefficient $>0.5$ and $p$-adjusted value$<0.05$). Finally, to identify regulatory links between accessible regions and active genes, peak to gene linkages were inferred using correlation analysis between enhancer accessibility and gene activity scores (*Granja et al., 2021*).

## Statistical analysis

All experiments were performed at least three times. Data are presented as mean ± SE. Multiple *t*-tests (recommended corrections), Student's *t*-test (parametric, unpaired, two-sided), or two-way

ANOVA were used for the evaluation of statistical significance using GraphPad 6 software. Statistical significance is presented as follows: *$p<0.05$, **$p<0.01$, ***$p<0.001$, **** $p<0.0001$.

## Study approval

All experiments were approved by the Institutional Committee of Protocol Evaluation in conjunction with the Veterinary Service Management of the Hellenic Republic Prefecture of Attika according to all current European and national legislation under the reference numbers: 4381- 07/07/2014 and 142752/22-02-2021 (EL 09 BIO exp 05) and comply to the ARRIVE guidelines.

## Materials and correspondence

Material requests and correspondence should be addressed to GK.

## Acknowledgements

We thank Dr. Aikaterini Nanou for critically reviewing the manuscript. We also thank Lida Iliopoulou, Anna Katevaini, Spiros Lalos, Panos Athanasakis, Michalis Meletiou, and Kleopatra Dagla for excellent technical assistance. We would also like to thank Fleming's animal house, flow cytometry (Sofia Grammenoudi), genomics (Vaggelis Harokopos), and microCT facilities.

This work was supported by the IMI project BTCure (GA no. 115142-2) to GK. We also acknowledge support of this work by Research Infrastructures InfrafrontierGR (MIS 5002135), which provided mouse hosting and phenotyping facilities, and pMedGR (MIS 5002802), which provided support for NGS experiments, funded by the Operational Programme 'Competitiveness, Entrepreneurship and Innovation' (NSRF 2014-2020), as well as project MIS 6004752 funded by the Regional Operational Programme 'ATTICA' (NSRF 2021-2027), co-financed by Greece and the European Union (European Reginal Development Fund). Lastly, we acknowledge support of this work by project SingleOut (HFRI-FM17C3-3780) funded by the Hellenic Foundation for Research and Innovation (HFRI) under the "1st Call for HFRI Research Projects to support Faculty members and Researchers and the procurement of high-cost research equipment", as well as by Horizon Europe Advanced ERC project BecomingCausal (ERC-2021-ADG, ID# 101055093) to GK.

## Additional information

### Competing interests

Niki Karagianni, Maria C Denis: affiliated with Biomedcode Hellas SA. The author has no financial interests to declare. The other authors declare that no competing interests exist.

### Funding

| Funder | Grant reference number | Author |
|---|---|---|
| Innovative Medicines Initiative | 115142-2 BTCure | George Kollias |
| Operational Programme "Competitiveness, Entrepreneurship and Innovation", NSRF 2014-2020, ERDF, EU/Greece | MIS 5002135 InfrafrontierGR | George Kollias |
| Regional Operational Programme "ATTICA" (NSRF 2021-2027), ERDF, Greece/EU | MIS 6004752 | George Kollias |
| Hellenic Foundation for Research and Innovation | HFRI-FM17C3-3780, SingleOut | George Kollias |
| HORIZON EUROPE European Research Council | 10.3030/101055093 | George Kollias |

| Funder | Grant reference number | Author |
|---|---|---|
| Operational Programme "Competitiveness, Entrepreneurship and Innovation", NSRF 2014-2020, ERDF, EU/Greece | MIS 5002802 pMedGR | George Kollias |

The funders had no role in study design, data collection and interpretation, or the decision to submit the work for publication.

## Author contributions

Fani Roumelioti, Conceptualization, Formal analysis, Validation, Investigation, Visualization, Methodology, Writing – original draft; Christos Tzaferis, Dimitris Konstantopoulos, Data curation, Software, Formal analysis, Validation, Visualization, Methodology, Writing – original draft; Dimitra Papadopoulou, Maria Sakkou, Anastasios Liakos, Theodore Meletakos, Yiannis Pandis, Validation, Investigation; Alejandro Prados, Validation, Methodology; Panagiotis Chouvardas, Software, Formal analysis, Validation; Niki Karagianni, Maria C Denis, Supervision; Maria Fousteri, Maria Armaka, Supervision, Validation; George Kollias, Conceptualization, Funding acquisition, Methodology, Project administration, Resources, Supervision, Validation, Writing – review and editing

## Author ORCIDs

Fani Roumelioti ⑩ http://orcid.org/0000-0003-0112-6601
Maria Armaka ⑩ https://orcid.org/0000-0003-0985-9076
George Kollias ⑩ https://orcid.org/0000-0003-1867-3150

## Ethics

All experiments were approved by the Institutional Committee of Protocol Evaluation in conjunction with the Veterinary Service Management of the Hellenic Republic Prefecture of Attika according to all current European and national legislation under the Reference numbers: 4381-07/07/2014 and 142752/22-02-2021 (EL 09 BIO exp 05) and comply to the ARRIVE guidelines.

## Decision letter and Author response

Decision letter https://doi.org/10.7554/eLife.84698.sa1
Author response https://doi.org/10.7554/eLife.84698.sa2

# Additional files

## Supplementary files

• Supplementary file 1. Primers used for mouse genotyping and qPCR primer sequences (5′–3′) are provided in (a and b).

• MDAR checklist

## Data availability

RNA-seq data are publicly available in the Gene Expression Omnibus (GEO) repository under the accession number GSE210887. Regarding the sc-ATAC seq data are already publicly available as part of *Armaka et al., 2022* (available here). Scripts/code for bioinformatic data analysis are uploaded in a public GitHub repository (copy archived at *Tzaferis, 2022*).

The following dataset was generated:

| Author(s) | Year | Dataset title | Dataset URL | Database and Identifier |
|---|---|---|---|---|
| Roumelioti F, Tzaferis C, Konstantopoulos D, Papadopoulou D, Prados A, Sakkou M, Liakos A, Chouvardas P, Meletakos T, Pandis Y, Karagianni N, Denis M, Fousteri M, Armaka M, Kollias G | 2022 | miR-221/222 drive synovial fibroblast expansion and pathogenesis of TNF-mediated arthritis | https://www.ncbi.nlm.nih.gov/geo/query/acc.cgi?acc=GSE210887 | NCBI Gene Expression Omnibus, GSE210887 |

The following previously published dataset was used:

| Author(s) | Year | Dataset title | Dataset URL | Database and Identifier |
|---|---|---|---|---|
| Armaka M, Konstantopoulos D, Tzaferis C, Lavigne MD, Sakkou M, Liakos A, Sfikakis P, Dimopoulos MA, Fousteri M, Kollias G | 2021 | Single-cell chromatin and transcriptome dynamics of Synovial Fibroblasts transitioning from homeostasis to pathology in TNF-driven arthritis | https://www.ncbi.nlm.nih.gov/bioproject/PRJNA778928 | NCBI BioProject, PRJNA778928 |

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
