## [Editor Report]

The findings of this work are important and offer significant advances to current knowledge. This manuscript used state of the art techniques and employed relevant animal models to provide convincing evidence supporting the regulatory role of microRNA cluster 221/222 in rheumatoid arthritis synovial fibroblast. Targeting miR-221/222 in SFs of patients harboring inflammatory arthritis might have a therapeutic benefit, and will be interesting to a wide range audience in the rheumatology and bone research fields.

---

## [Decision Letter]

**Decision letter after peer review:**

Thank you for submitting your article "miR-221/222 drive synovial hyperplasia and arthritis by targeting cell cycle inhibitors and chromatin remodeling components" for consideration by *eLife*. Your article has been reviewed by 3 peer reviewers, one of whom is a member of our Board of Reviewing Editors, and the evaluation has been overseen by Mone Zaidi as the Senior Editor. The following individual involved in the review of your submission has agreed to reveal their identity: Gerasimos Evangelatos (Reviewer #3).

Essential revisions (for the authors):

1) Figure 3: the model in which miR221/222 are overexpressed in SFs does not lend itself to examining potential effects on other cell populations other than indirect effects. It would be informative to compare with overexpression in immune cells, especially in myeloid cells that give rise to osteoclasts to determine the direct effect on bone erosion

2) Although the focus of this study is largely on miR221/222 regulation of SFs, the observations that deletion of these miRs protects against bone erosion (less OCs) yet does not regulate inflammatory influx (Figure S8), begs the question, is the effect on myeloid progenitors direct or SF-mediated? The global deletion of miRs and restricting the analysis to SFs limits better understanding and appreciation of the mechanism in joint pathology.

3) Given the significant implication of this study for the potential modulation of inflammatory arthritis, it would be of great interest to determine if miR221/222 knockout are protected in CIA or CAIA experimental models of RA (in addition to the huTNFtg model).

4) In figure 3, although a significant increase in CD8 subpopulation is observed, its possible role in the pathogenesis of disease is not clearly discussed in the manuscript. Increased levels of CD8(+) T cells have been described in RA doi.org/10.1002/art.38941. Moreover, miR221/222 directly target CD4 doi: 10.1016/j.celrep.2017.09.030. All these studies suggest an implication of the immune system in miR221/222 mediated RA pathogenesis. Further characterization of the immune panel should be provided, analyzing specific CD8 subpopulations and their role in disease progression should be tested by using immunodepletion in huTNFtg and huTNFtg miR221/222 mice.

5) Figure 4 and 5 The expression of genes modulated by miR221/222 is shown only as an mRNA; also protein expression should be analysed.

6) Lines 122-129: Do the authors have any data about TNF expression in TgColVI-miR-221/222 mouse, especially in their SFs? Given the defective bone physiology demonstrated later in these mice, it would be interesting to know if they also exert a proinflammatory dynamic

*Reviewer #1 (Recommendations for the authors):*

Figure 3: the model in which miR221/222 are overexpressed in SFs does not lend itself to examining potential effects on other cell populations other than indirect effects. It would be informative to compare with overexpression in immune cells, especially in myeloid cells that give rise to osteoclasts to determine the direct effect on bone erosion

Although the focus of this study is largely on miR221/222 regulation of SFs, the observations that deletion of these miRs protects against bone erosion (less OCs) yet does not regulate inflammatory influx (Figure S8), begs the question, is the effect on myeloid progenitors direct or SF-mediated? The global deletion of miRs and restricting the analysis to SFs limits better understanding and appreciation of the mechanism in joint pathology.

It would be more informative and would add additional rigor to current data if well-established markers of destructive lining SFs are documented in this model.

The study design (mouse models) approaches miR221/222 as a cluster dismissing some divergent expressions of the individual miRs as apparent in some figures. Do the authors consider this point insignificant or irrelevant? If so, please provide supporting evidence/argument.

Given the significant implication of this study for the potential modulation of inflammatory arthritis, it would be of great interest to determine if miR221/222 knockout is protected in CIA or CAIA experimental models of RA (in addition to the huTNFtg model).

The statement that miR221/222 may be implicated in bone physiology is an overstatement and potentially misleading. The reported changes in bone parameters could simply be the result of secondary effects of miR221/222 abnormal overexpression, which we should recognize is global. These statements should be carefully reconsidered.

*Reviewer #2 (Recommendations for the authors):*

Figure 1 – Although miR221 and miR222 are part of the same genetic locus, the regulation of their expression seems to be different. For instance, as shown in Figure 1CD, mirR221 is significantly downregulated by polyIC, whereas miR222 is upregulated. Also, miR222 seems to be more expressed in ex vivo SFs at least in the first weeks (Figure 1AB). Please, comment on this.

In figure 3, although a significant increase in CD8 subpopulation is observed, its possible role in the pathogenesis of disease is not clearly discussed in the manuscript. Increased levels of CD8(+) T cells have been described in RA doi.org/10.1002/art.38941. Moreover, miR221/222 directly target CD4 doi: 10.1016/j.celrep.2017.09.030. All these studies suggest an implication of the immune system in miR221/222 mediated RA pathogenesis. Further characterization of the immune panel should be provided, analyzing specific CD8 subpopulations and their role in disease progression should be tested by using immunodepletion in huTNFtg and huTNFtg miR221/222 mice.

Figures 4 and 5 -The expression of genes modulated by miR221/222 is shown only as an mRNA; also protein expression should be analysed. Moreover, the authors identified Smarca1 as a novel predicted miR-221/222 target. According to Targetscan, Smarca1 is the target of miR222 but not miR221. Please, clarify. Moreover, the direct binding should be validated by luciferase assay.

Methods – The information on the statistical test used should be implemented. Since different types of experiments are present in the study (e.g. paired, unpaired samples) and the authors state that they use different t-tests (parametric, unpaired, two-sided) or two-way ANOVA, which test is used for each experiment should be reported.

*Reviewer #3 (Recommendations for the authors):*

Lines 56, 62, 257-258: A useful review that could be also included is "MicroRNAs in rheumatoid arthritis: From pathogenesis to clinical impact. Evangelatos G, Fragoulis GE, Koulouri V, Lambrou GI. Autoimmun Rev. 2019 Nov;18(11):102391. doi: 10.1016/j.autrev.2019.102391".

Lines 26, 27, 79, 87, 104, 110, 123, etc: Please be consistent throughout the text, by using either "microRNAs" or the abbreviation "miRNAs", not both.

Lines 87-88 "in specific subpopulations": The authors should explain what they exactly mean; do they refer to a "specific cell subpopulation" generally or to "specific SF subpopulations"?

Line 115: To which figure does "Figure EV1C" refer?

Lines 122-129: Do the authors have any data about TNF expression in TgColVI-miR-221/222 mouse, especially in their SFs? Given the defective bone physiology demonstrated later in these mice, it would be interesting to know if they also exert a proinflammatory dynamic.

Lines 141-142, "did not lead to a significant increase of inflammatory infiltrations, apart from an increase in CD8^+^ T cells": The authors could possibly add "compared to huTNFtg mouse".

Lines 157-159, "both WT and huTNFtg SFs … compared to their respective controls (Figure 3E-F)": I feel that the message of this sentence is not exactly what is illustrated in the relative figure. It should be better if the authors further explain this point.

Line 187, "as well as in SFs from huTNFtg mice compared to SFs from WT mice": How does this support that p27, p57, and Smarca1 are targets or miR-221/222? Did the authors possibly mean that the expression of p27, p57, and Smarca1 was downregulated in TgColVI-miR-221/222 compared to huTNFtg?

Lines 330-332: The authors should recognize that they examined the therapeutic effect of miR-221/222 inhibition only by generating a miR-221/222 -/- knock-out mouse and not by applying a miR-221/222 inhibitor in huTNFtg, TgColVI-miR-221/222 or huTNFtg;TgColVI-miR-221/222 mice.

---

## [Author Response]

Essential revisions (for the authors):1) Figure 3: the model in which miR221/222 are overexpressed in SFs does not lend itself to examining potential effects on other cell populations other than indirect effects. It would be informative to compare with overexpression in immune cells, especially in myeloid cells that give rise to osteoclasts to determine the direct effect on bone erosion2) Although the focus of this study is largely on miR221/222 regulation of SFs, the observations that deletion of these miRs protects against bone erosion (less OCs) yet does not regulate inflammatory influx (Figure S8), begs the question, is the effect on myeloid progenitors direct or SF-mediated? The global deletion of miRs and restricting the analysis to SFs limits better understanding and appreciation of the mechanism in joint pathology.

We thank the reviewer for these interesting comments. We have now crossed the huTNFtg;TgCol6a1-Mir221/222 mice in a Mir221/222 ^-/-^ background, so as to check the SF-specific role of *Mir221/222* in arthritis manifestations. As depicted in updated Figure 2G, we can observe that huTNFtg;TgCol6a1-Mir221/222;Mir221/222 ^-/-^ mice (where *Mir221/222* are only expressed in SFs and are absent in other cell types) exhibited the same arthritic manifestations as the huTNFtg;TgCol6a1-Mir221/222;Mir221/222 ^d/f,f/-^ mice, as such include synovial hyperplasia, cartilage destruction and number of osteoclasts. This indicates that SF-specific *Mir221/222* overexpression is sufficient for potentiation of arthritis progression even when *Mir221/222* expression is missing from other cell types.

We have presented these new findings in Figure 2G and lines 146-158 in the text.

3) Given the significant implication of this study for the potential modulation of inflammatory arthritis, it would be of great interest to determine if miR221/222 knockout are protected in CIA or CAIA experimental models of RA (in addition to the huTNFtg model).

We performed a CAIA experiment in Mir221/222 ^-/-^ (ko) and Mir221/222 ^f/-^ littermate male mice. Interestingly, a trend towards disease amelioration was observed in Mir221/222 ^-/-^ mice resembling the effect in the huTNFtg mouse model. Indeed, a larger number of mice would strengthen statistics, however this has not been possible within the given time limits for resubmission, due to technical issues with the matings. This is not unexpected as both models depend on TNF signals, be it acute (CAIA) or chronic (huTNFtg). Experimental findings (clinical score and weight) are presented in the following graphs and available in the peer-review process but not included in the final version of the paper.

**Author response image 1. sa2fig1:** (A) Clinical score in CAIA induced arthritis in Mir221/222 -/- (ko) and Mir221/222 f/- littermate male mice. (B) Weight measurements of Mir221/222 ^-/-^ (ko) and Mir221/222 ^f/-^ littermate male mice during the CAIA induced arthritis protocol.

4) In figure 3, although a significant increase in CD8 subpopulation is observed, its possible role in the pathogenesis of disease is not clearly discussed in the manuscript. Increased levels of CD8(+) T cells have been described in RA doi.org/10.1002/art.38941. Moreover, miR221/222 directly target CD4 doi: 10.1016/j.celrep.2017.09.030. All these studies suggest an implication of the immune system in miR221/222 mediated RA pathogenesis. Further characterization of the immune panel should be provided, analyzing specific CD8 subpopulations and their role in disease progression should be tested by using immunodepletion in huTNFtg and huTNFtg miR221/222 mice.

We have now characterized the CD8^+^ T cell subpopulations in the joints of huTNFtg and huTNFtg;TgCol6a1-Mir221/222 mice and effector CD8^+^ T cells is the predominant CD8^+^ T cell subpopulation, that is also found increased in huTNFtg;TgCol6a1-Mir221/222 mice compared to the huTNFtg. The gating strategy and the new results are now presented in Figure 3—figure supplement 1B and Figure 3—figure supplement 2A-D and lines 154-158 in the text. We did not proceed to CD8^+^ immunodepletion as TNF-induced arthritis in the huTNFtg mouse model fully develops in the absence of mature T and B cells (*Kollias et al., Immunol. Rev 1999*, DOI: 10.1111/j.1600-065x.1999.tb01315.x).

5) Figure 4 and 5 The expression of genes modulated by miR221/222 is shown only as an mRNA; also protein expression should be analysed.

We analyzed the protein expression of CDKN1B (P27), CDKN1C(P57) and SMARCA1 and found that also the protein levels of these targets are downregulated upon *Mir221/222* overexpression and upregulated when *Mir221/222* are deleted in arthritogenic SFs. We have now included these new results in Figure 4H-L, Figure 5G-K and lines 214-218 and 266-268 in the main text.

6) Lines 122-129: Do the authors have any data about TNF expression in TgColVI-miR-221/222 mouse, especially in their SFs? Given the defective bone physiology demonstrated later in these mice, it would be interesting to know if they also exert a proinflammatory dynamic

Indeed, we have now measured the human TNF expression in supernatants from SFs of the huTNFtg;TgCol6a1-Mir221/222 and huTNFtg mice and no difference was observed. Moreover, we have included human TNF expression data from the sera of the aforementioned mice showing again no difference between the huTNFtg;TgCol6a1-Mir221/222 and huTNFtg mice. All new results are now presented in Figure 4—figure supplement 2F,G and lines 234-243 in the text.

Reviewer #1 (Recommendations for the authors):It would be more informative and would add additional rigor to current data if well-established markers of destructive lining SFs are documented in this model.

We thank the reviewer for this interesting comment. Our RNA expression data from cultured SFs from huTNFtg;TgCol6a1-Mir221/222 mice compared to the huTNFtg did not reveal differences in genes linked with a destructive identity of lining SFs, but rather in genes implicated in cell cycle progression, proliferation and extracellular matrix remodeling (Excel file: Figure_4A-B_and_Figure_4—figure supplement_1_Source data 1). We had validated the expression of genes related to the extracellular matrix, that could be linked to a destructive lining identity of SFs, such as *Prg4* and *Itga3* (Figure 4F,G) and discussed the results in the Discussion section (lines 330-351). Additionally, RNA expression levels of other genes implicated in a destructive behavior of SFs, such as *Mmp3* and *Rankl* (*Tnfsf11*), were not found to differ in arthritogenic SFs upon *Mir221/222* overexpression (revised Figure 4—figure supplement 2A,B and lines 225-229 in the main text).

The study design (mouse models) approaches miR221/222 as a cluster dismissing some divergent expressions of the individual miRs as apparent in some figures. Do the authors consider this point insignificant or irrelevant? If so, please provide supporting evidence/argument.

Indeed, there are small differences in expression of *Mir221* and *Mir222* (Figure 1C,D). However, both miRNAs show similar upregulation in arthritogenic SFs (Figure 1A, B) and under TNF and IL-1b stimulation (Figure 1C,D) and share common targets. Future studies exploiting additional tools could shed light on the individual role of these miRs in arthritis.

The statement that miR221/222 may be implicated in bone physiology is an overstatement and potentially misleading. The reported changes in bone parameters could simply be the result of secondary effects of miR221/222 abnormal overexpression, which we should recognize is global. These statements should be carefully reconsidered.

We appreciate this comment and we have rephrased the suggestion in the text in lines 142-146. We would also like to mention that we refer to a potential function of these two miRNAs in bone physiology pointing out that it needs further experimental elucidation.

Reviewer #2 (Recommendations for the authors):Figure 1 – Although miR221 and miR222 are part of the same genetic locus, the regulation of their expression seems to be different. For instance, as shown in Figure 1CD, mirR221 is significantly downregulated by polyIC, whereas miR222 is upregulated. Also, miR222 seems to be more expressed in ex vivo SFs at least in the first weeks (Figure 1AB). Please, comment on this.

We thank the reviewer for this interesting observation. *Mir221* and *Mir222* expression is reported to be controlled by the same promoter and enhancer elements (24). We are not aware and it would be interesting for a future study to dissect if post-transcriptional mechanisms exist that also modify the expression of these two miRNAs under specific signals. However, inflammatory signals that exist in the arthritic synovium such as TNF and IL-1b appear to control *Mir221* and *Mir222* expression similarly.

Figures 4 and 5 -The expression of genes modulated by miR221/222 is shown only as an mRNA; also protein expression should be analysed. Moreover, the authors identified Smarca1 as a novel predicted miR-221/222 target. According to Targetscan, Smarca1 is the target of miR222 but not miR221. Please, clarify. Moreover, the direct binding should be validated by luciferase assay.

We agree with the reviewer that protein expression should be analyzed. We analyzed the protein expression of CDKN1B (P27), CDKN1C(P57) and SMARCA1 and found that also the protein levels of these targets are downregulated upon *Mir221/222* overexpression and upregulated when *Mir221/222* are deleted in arthritogenic SFs. We have now included these new results in Figure 4H-L, Figure 5G-K and lines 214-218 and 266-268 in the main text.

We apologize for not having made it clearer in the manuscript. *Smarca1* was predicted to be a target for *Mir221* and *Mir222* by DIANA-microT-CDS and Pictar. We have clarified it now in the lines 207-209 in the revised manuscript.

Indeed, the point of the reviewer referring to the validation of *Smarca1* as a direct target of *Mir221/222* by a luciferase assay is important. We have now performed the assay and *smarca1* is found to be a direct target of *Mir221/222*. The new result can be found in Figure 5L and in lines 84, 271-276 and 352-354 in the text.

Methods – The information on the statistical test used should be implemented. Since different types of experiments are present in the study (e.g. paired, unpaired samples) and the authors state that they use different t-tests (parametric, unpaired, two-sided) or two-way ANOVA, which test is used for each experiment should be reported.

We have now updated the suggested corrections and corrected the statistical method used in the figure legends. We also referred to the statistical methods used in the Materials and methods-Statistical analysis section.

Reviewer #3 (Recommendations for the authors):Lines 56, 62, 257-258: A useful review that could be also included is "MicroRNAs in rheumatoid arthritis: From pathogenesis to clinical impact. Evangelatos G, Fragoulis GE, Koulouri V, Lambrou GI. Autoimmun Rev. 2019 Nov;18(11):102391. doi: 10.1016/j.autrev.2019.102391".

We thank the reviewer for this suggestion and we refer to the review in the revised manuscript.

Lines 26, 27, 79, 87, 104, 110, 123, etc: Please be consistent throughout the text, by using either "microRNAs" or the abbreviation "miRNAs", not both.

We have now corrected it in the revised manuscript.

Lines 87-88 "in specific subpopulations": The authors should explain what they exactly mean; do they refer to a "specific cell subpopulation" generally or to "specific SF subpopulations"?

We appreciate this comment. We mean specific SF subpopulations and it is corrected in line 90 of the manuscript.

Line 115: To which figure does "Figure EV1C" refer?

It refers to Figure 1—figure supplement 1C and we have now corrected it in line 120.

Lines 141-142, "did not lead to a significant increase of inflammatory infiltrations, apart from an increase in CD8^+^ T cells": The authors could possibly add "compared to huTNFtg mouse".

We have added the reviewer’s suggestion in line 162.

Lines 157-159, "both WT and huTNFtg SFs … compared to their respective controls (Figure 3E-F)": I feel that the message of this sentence is not exactly what is illustrated in the relative figure. It should be better if the authors further explain this point.

By stating this in lines 182-186 (revised manuscript) we refer to the pro-proliferative and wound-healing promoting role of *Mir221/222* in SFs in homeostatic and disease conditions.

Line 187, "as well as in SFs from huTNFtg mice compared to SFs from WT mice": How does this support that p27, p57, and Smarca1 are targets or miR-221/222? Did the authors possibly mean that the expression of p27, p57, and Smarca1 was downregulated in TgColVI-miR-221/222 compared to huTNFtg?

We have now revised this statement in line 216-217 based on the generated protein expression data of the targets.

Lines 330-332: The authors should recognize that they examined the therapeutic effect of miR-221/222 inhibition only by generating a miR-221/222 -/- knock-out mouse and not by applying a miR-221/222 inhibitor in huTNFtg, TgColVI-miR-221/222 or huTNFtg;TgColVI-miR-221/222 mice.

We have now stated “genetic deletion” in lines 376-377 and we refer to future therapeutic targeting of these two miRNAs that could be beneficial in RA.